# Universal Algorithm-Implicit Learning

Stefano Woerner [1]   Seong Joon Oh [2]   Christian F. Baumgartner [1 3]

## Abstract

Current meta-learning methods are constrained to narrow task distributions with fixed feature and label spaces, limiting applicability. Moreover, the current meta-learning literature uses key terms like "universal" and "general-purpose" inconsistently and lacks precise definitions, hindering comparability. We introduce a theoretical framework for meta-learning which formally defines *practical universality* and introduces a distinction between *algorithm-explicit* and *algorithm-implicit* learning, providing a principled vocabulary for reasoning about universal meta-learning methods. Guided by this framework, we present TAIL, a transformer-based algorithm-implicit meta-learner that functions across tasks with varying domains, modalities, and label configurations. TAIL features three innovations over prior transformer-based meta-learners: random projections for cross-modal feature encoding, random injection label embeddings that extrapolate to larger label spaces, and efficient inline query processing. TAIL achieves state-of-the-art performance on standard few-shot benchmarks while generalizing to unseen domains. Unlike other meta-learning methods, it also generalizes to unseen modalities, solving text and audio classification tasks despite training exclusively on images, handles tasks with up to $20\times$ more classes than seen during training, and provides orders-of-magnitude computational savings over prior transformer-based approaches.

## 1. Introduction

Modern deep learning has achieved remarkable success by leveraging large datasets and heavy computation. How-ever, in many real-world settings, collecting large labeled datasets is costly, ethically constrained, or infeasible. Meta-learning, or "learning to learn", addresses this challenge by training algorithms which can rapidly adapt to new tasks from only a few examples. Despite this promise, existing meta-learning methods still struggle to learn strategies that transfer robustly across diverse task types, which require the meta-learner to develop generalizable learning strategies rather than overfitting to domain-specific features. We ar-gue that a true "learning-to-learn" algorithm should succeed even when the target domain is entirely different from the source and propose that meta-learning can benefit from a paradigm shift analogous to that of deep learning by exploiting large-scale *meta*-datasets.

Recently, it has been shown that many meta-learning algo-rithms are limited when there is a large shift between the feature domain of the meta-training data and the applica-tion dataset (Chen et al., 2020; Luo et al., 2023; Guo et al., 2020; Oh et al., 2022; Hospedales et al., 2021; Vettoruzzo et al., 2024). We hypothesize that most existing methods suffer from structural limitations which prevent such broad generalization. However, there is currently no theoretical framework for meta-learning that provides the right taxon-omy to surface such structural limitations. Moreover, the meta-learning literature lacks precise definitions for key terms like "universal" and "general-purpose", which are being used inconsistently in different works. This termino-logical ambiguity hinders comparability and obscures the true level of generalization which a method can achieve.

To address these issues, we first introduce a theoretical framework for meta-learning, which allows us to formally describe important properties of meta-learning paradigms. Within this framework, we propose a distinction between *algorithm-explicit* and *algorithm-implicit* learning systems, which proves crucial for understanding why some meta-learning approaches generalize more broadly than others. Moreover, we define the notion of *practical universality*, describing the property of functioning as a robust learn-ing algorithm across diverse task distributions that vary in feature domains, label spaces, and loss functions.

Building on this framework, we present a novel *algorithm-implicit* meta-learning method called TAIL that makes substantial advances towards practical universality (see

---

[1]Cluster of Excellence "Machine Learning: New Perspectives for Science", University of Tübingen, Germany [2]Tübingen AI Cen-ter, University of Tübingen, Germany [3]Faculty of Health Sciences and Medicine, University of Lucerne, Switzerland. Correspon-dence to: Stefano Woerner <stefano@woerner.eu>.

*Proceedings of the $43^{rd}$ International Conference on Machine Learning*, Seoul, South Korea. PMLR 306, 2026. Copyright 2026 by the author(s).

*Table 1.* We compare TAIL to other sequence-based meta-learners (SNAIL (Mishra et al., 2018), GPICL (Kirsch et al., 2024), CAML (Fifty et al., 2023)) in dimensions relevant to practical universality.

| Method | Causal model? | Variable feature spaces | Variable label spaces | Flexible seq. length | Key Limitation |
|---|---|---|---|---|---|
| SNAIL | Causal | ✗ | ✗ | ✗ | Cannot generalize across modalities, label spaces or support set size |
| GPICL | Causal | theoretically | ✗ | ✗ | Cannot generalize across label spaces or support set size; no cross-modality experiments |
| CAML | Non-causal | ✗ | ✗ | ✓ | Cannot generalize across modalities or label spaces |
| TAIL (ours) | Non-causal | ✓ | ✓ | ✓ | — |

Figure 1). While the theoretical framework is task-type-agnostic, we focus on classification as a first demonstration, where varying label spaces, domains, and modalities already represent a substantial advance over prior work. Following prior work (Santoro et al., 2016; Kaiser et al., 2017; Kirsch et al., 2024; Fifty et al., 2023), we reformulate the few-shot learning problem as a sequence modeling problem, using a transformer on sequences of data-label-pairs and an unlabeled query sample. Prior approaches were limited to toy datasets, single domains or single modalities and did not generalize well across domains (see Tab. 1). Guided by our theoretical framework, we address three key challenges that limit the practical universality of existing meta-learning methods. **(i) Universal feature handling:** We develop a feature encoding strategy that combines task-specific encoders with randomly sampled projections into a common latent space, enabling seamless transfer across completely different modalities (images, text, medical scans) without architectural modifications and without retraining. **(ii) Universal label handling:** We introduce a randomized global dictionary of learnable embeddings, allowing the model to handle arbitrary label sets and extrapolate to tasks with more classes than seen during training. **(iii) Computational efficiency:** Our transformer-based method scales to tasks with much larger label sets while maintaining strong performance and requiring only a fraction of the computation of previous transformer-based methods.

These innovations enable our algorithm to scale to training on large and diverse meta-datasets. Analogous to the transition in deep learning, where large datasets supplanted explicit feature engineering and strong hand-crafted inductive biases, the algorithm-implicit meta-learning paradigm has the potential to drive a similar shift, replacing explicitly defined learning procedures with models that implicitly learn how to learn from large-scale meta-datasets.

We demonstrate that our method achieves new state-of-the-art performance across a range of few-shot classification benchmarks and generalizes robustly to unseen domains and modalities. Notably, it maintains strong performance on tasks involving up to $20\times$ more classes than observed during training, a capability not exhibited by existing meta-learning approaches. Moreover, our method performs well on highly diverse domains not seen during training. Most strikingly, it successfully solves text-based and audio-based few-shot learning tasks despite being trained exclusively on image data.

**Code:** We make our code publicly available on GitHub.[1]

**Conflict of Interest Disclosure:** The authors declare no financial conflicts of interest.

## 2. Background and Notation

### 2.1. The Learning Problem

We formalize a learning task as $T := (\mathcal{X}, \mathcal{Y}, p, \ell)$, where $\mathcal{X}$ is the feature domain or input domain, $\mathcal{Y}$ is the label domain or output domain, $p(x, y)$ is a distribution of data with $x \in \mathcal{X}, y \in \mathcal{Y}$, and $\ell : \mathcal{Y} \times \mathcal{Y} \to \mathbb{R}$ is a loss function, which measures a "distance" between a predicted value and the ground truth label. $\ell$ is assumed to be measurable, non-negative, but not necessarily a true metric.

The learning problem on task $T$ consists of finding a function $f : \mathcal{X} \to \mathcal{Y}$, which is typically called "hypothesis" or "model", that minimizes the risk

$$R(f) = \mathop{\mathbb{E}}_{(x,y) \sim p(x,y)} [\ell(f(x), y)] .$$

In practice, the true data distribution is unknown, and $f$ is estimated using a sample from $p$ called the training set or support set $S$, using supervised learning.

Formally, given the space $\mathcal{D} := \mathrm{supp}(p) \subseteq \mathcal{X} \times \mathcal{Y}$ of possible data, defined by the support of the distribution $p$, we can define a support set $S \subset \mathcal{D}$ as $S := \{(x_i, y_i)\}_{i=1}^{|S|}$ with $(x_i, y_i) \in \mathrm{supp}(p)$. With this notation we can define:

**Definition 2.1** (Learning Algorithm). A learning algorithm $\mathcal{A} : \mathcal{P}(\mathcal{X} \times \mathcal{Y}) \to \mathcal{Y}^{\mathcal{X}}$, $S \mapsto f$ is a function that maps a dataset $S \subseteq \mathcal{D}$ to a hypothesis $f$.

### 2.2. Meta-Learning

Meta-learning can be understood as finding a learning algorithm $\mathcal{A}$ through meta-optimization over the space $\mathbb{A}$ of possible learning algorithms. In meta-learning we assume access to a meta-dataset $\mathcal{T}$ of tasks sampled from a distribution of tasks $\nu$. The meta-learning problem can now be described as finding a learning algorithm $\mathcal{A}$ that minimizes

$$R^{\mathrm{Meta}}(\mathcal{A}) = \mathop{\mathbb{E}}_{T \sim \nu} \mathop{\mathbb{E}}_{S \sim \bigcup_{n \geq 1} p_T^n} R_T(\mathcal{A}(S))$$

$$= \mathop{\mathbb{E}}_{T \sim \nu} \mathop{\mathbb{E}}_{S \sim \bigcup_{n \geq 1} p_T^n} \mathop{\mathbb{E}}_{(x,y) \sim p_T} [\ell_T(\mathcal{A}(S)(x), y)] .$$

---

[1] https://github.com/StefanoWoerner/TAIL

# 3. Universal Algorithm-Implicit Learning

Prior meta-learning literature lacks a unified theoretical framework for reasoning about generalization across domains, modalities, and label spaces. Different works use inconsistent terminology: terms like "universal", "general-purpose", and "cross-domain" are employed with varying meanings, often without precise definitions. For instance, some works claim "universality" while restricting to fixed feature dimensions (Fifty et al., 2023), others use "general-purpose" but evaluate only on single modalities (Kirsch et al., 2024). The lack of clear definitions makes different works difficult to compare and might lead to a mismatch between expectation and reality when a certain term is used.

Here, we provide a formal definition of universal learning and introduce the term of **practical universality**. Our definition adopts a broader notion of universality than prior work. It aims at describing algorithms which work on varying feature spaces, label spaces, and task types across different modalities and domains. Additionally, we propose a distinction **algorithm-explicit vs. algorithm-implicit** meta-learning paradigms, which clarifies why certain approaches generalize better than others. We believe that this distinction is pivotal for understanding meta-learning paradigms.

## 3.1. Demonstration-Conditioned Inference

Rather than first learning a hypothesis and using it for prediction, one can directly make predictions for a data point conditioned on a support set, without explicitly computing a hypothesis $f$. We refer to such a mapping as a *demonstration-conditioned inference (DCI) function*.

**Definition 3.1** (Demonstration-Conditioned Inference)**.** A DCI function $g : \mathcal{P}(\mathcal{X} \times \mathcal{Y}) \times \mathcal{X} \to \mathcal{Y}, (S, x) \mapsto \hat{y}$ is a function that directly maps a support set $S \subseteq \mathcal{D}$ and query point $x$ to a prediction $\hat{y}$.

Note that for any deterministic DCI function $g$ and fixed $S$, there exists an induced hypothesis $f_S$ where $f_S(x) = g(S, x)$. That is, $g$ implicitly defines a learning algorithm $\mathcal{A}_g$ where $\mathcal{A}_g(S) = f_S$. Vice-versa, an algorithm $\mathcal{A}$ induces a DCI $g_\mathcal{A}$ with $g_\mathcal{A}(S, x) = \mathcal{A}(S)(x)$.

## 3.2. Algorithm-Explicit vs. Algorithm-Implicit Learning

We introduce a fundamental distinction between learning paradigms based on whether the learning algorithm is explicitly specified or implicitly emerges.

**Algorithm-Explicit Learning:** Intuitively, a learning system is *algorithm-explicit* if its training procedure is explicitly specified. We formally define such a system, as one which is characterized by an explicit procedure $\mathcal{A}$ that maps a dataset $S$ to a hypothesis $f$ (or equivalently $g$), i.e. $\mathcal{A}(S) = f$. An example for algorithm-explicit learning are

MLPs optimized by (stochastic) gradient descent. The learning algorithm $\mathcal{A}$ on a training set $S$ is defined as iteratively updating the MLP's parameters $\theta$ with GD steps on samples from $S$, yielding the hypothesis $f_\theta$. Other examples of explicitly defined learning systems are $k$-nearest neighbors (Cover & Hart, 1967) ($\mathcal{A}$ stores $S$ and $f$ performs distance-based voting), or MAML (Finn et al., 2017) ($\mathcal{A}$ performs $k$ steps of gradient descent from initialization $\theta_0$ to find $f$).

**Algorithm-Implicit Learning:** A learning system is *algorithm-implicit* if it operates through a parameterized DCI function $g_\theta$ where the learning algorithm $\mathcal{A}_g$ emerges from the learned parameters $\theta$ but is never explicitly specified. The implicit $\mathcal{A}_g$ is defined only through the behavior of $g_\theta(S, \cdot)$ for various $S$. This means that implicit learning algorithms $\mathcal{A}$ are black boxes with minimal inductive biases. Examples of algorithm-implicit learning are attention-based meta-learners, such as SNAIL (Mishra et al., 2018), CAML (Fifty et al., 2023) and GPICL (Kirsch et al., 2024), or in-context learning methods (Brown et al., 2020; Wu et al., 2025).

Algorithm-explicit methods have externally specified rules and may only learn a narrow set of parameters. These methods with explicitly defined learning procedures embed assumptions about the structure of optimal solutions. The resulting strong inductive biases help when meta-training data are scarce but become liabilities when target tasks differ substantially from training tasks. In contrast, algorithm-implicit approaches place no assumptions on the learning algorithm, allowing the model to task-distribution-specific algorithms that adapt to the structure present in diverse meta-training data, but requiring more meta-training tasks to do so. Any inductive biases come only from the computational structure of $g$, not from explicit design constraints of the learning algorithm. Our method, which we describe in Section 4, follows an algorithm-implicit approach.

## 3.3. Practical Universality and Universal Learning Algorithms

Prior work does not provide a formal definition that captures what a truly universal few-shot learner should achieve. In the following, we derive *practical universality* as a formal criterion that addresses this gap.

Traditional learning theory provides notions of universality that are asymptotic in nature, but these are insufficient to make statements about few-shot learning. The classical notion of *universal consistency* guarantees convergence to Bayes-optimal performance as sample size grows to infinity, but makes no claims about finite-sample behavior.

**Definition 3.2** (Universal Consistency)**.** A learning algorithm $\mathcal{A}$ is *consistent* w.r.t. a distribution $p$ over $\mathcal{X} \times \mathcal{Y}$ if the risk of the model $\mathcal{A}(S_n)$ converges to the Bayes risk $R^* = \inf_f R(f)$, as the size of the support set $n = |S_n| \to$

$\infty$ where $S_n \sim p^n$, i.e. if $\lim_{n \to \infty} R(\mathcal{A}(S_n)) = R^*$.
A learning algorithm $\mathcal{A}$ is *universally consistent* if it is consistent for any distribution $p$.

To be able to analyze the finite sample performance of an algorithm, we formalize the idea of a *learning curve*, which describes how the expected performance of an algorithm evolves as a function of the number of training samples. The term is used in various works in statistical learning theory with equivalent or similar definitions (Schuurmans, 1997; Viering & Loog, 2023).

**Definition 3.3** (Learning Curve). The learning curve

$$\alpha_T(\mathcal{A}, n) = \mathop{\mathbb{E}}_{S \sim p_T^n} [R_T(\mathcal{A}(S)) - R_T^*].$$

of an algorithm $\mathcal{A}$ on a task $T$ computes the expected residual risk conditioned on the size of the support set.

Intuitively, we would like a learning algorithm to have a lower residual risk with an increasing amount of training data. That is, a valid learning algorithm should, on average, not perform worse when given more data. With this, we can define the notions of a *valid learning algorithm* and of *universal validity*, analogous to the asymptotic notions of consistency and universal consistency.

**Definition 3.4** (Valid Learning Algorithm). An algorithm $\mathcal{A}$ qualifies as a *valid learning algorithm* for task $T$ if $\alpha_T(\mathcal{A}, n)$ is monotonically non-increasing in $n$ and for any $\varepsilon > 0$ there exists an $n$ with $\alpha_T(\mathcal{A}, n) < \varepsilon$.

**Definition 3.5** (Universal Validity). $\mathcal{A}$ is universally valid w.r.t. a distribution of tasks $\nu$ if it is a valid learning algorithm for all tasks $T$ in the class of tasks $\text{supp}(\nu)$.

**Practical Universality:** Note that we defined universal validity as a property of a learning algorithm *with respect to* a distribution of tasks. Typically, in meta-learning research, all tasks in the meta-dataset $T \in \mathcal{T}$ are considered to have the same feature space, label space and loss function. Most papers implicitly assume that $\mathcal{Y}_T \cong \{1, \cdots, K\}$ for a fixed number of classes $K$. Instead, here we allow tasks to have feature spaces, label spaces and data distributions that differ from each other. Importantly, a test time task $T' \in \mathcal{T}_{\text{test}}$ might have a feature space or label space that is not represented in $\mathcal{T}_{\text{train}}$. We classify an algorithm $\mathcal{A}$ (or a DCI $g_\theta$ inducing an algorithm $\mathcal{A}_g$) as a **practically universal** learning algorithm only if it is a universally valid learning algorithm on a distribution containing such test time tasks.

### 3.4. Few-Shot Benchmarking

Intuitively, a good few-shot algorithm needs to learn a new task using only a limited number of labeled examples. In practice, tasks are presented as a pair $(S, Q)$ of support and query sets and not with their underlying data distribution.

For each *episode* of $N$-shot learning on a task $T$, we sample a support set $S \subset \mathcal{D}_T = \text{supp}(p_T)$ such that $S$ contains $N$ i.i.d. samples from $p_T(x \mid y)$ for each label $y \in \mathcal{Y}$ in the task. We then sample a query set $Q \subset \mathcal{D} \setminus S$ such that it contains a fixed number $N_Q$ of i.i.d. query samples from $p_T(x \mid y)$ for each label $y \in \mathcal{Y}$ in the task. We write $S \sim p_T^n$ and $Q \sim p_T^{n'}$ with $n := N \cdot |\mathcal{Y}| = |S|$ and $n' := N_Q \cdot |\mathcal{Y}| = |Q|$. We call the resulting pair $(S, Q)$ an $N$-shot instance of task $T$. Evaluating algorithms on many $N$-shot episodes sampled from our distribution of tasks can is used to estimate $\mathbb{E}_{T \sim \nu} \alpha_T(\mathcal{A}, n)$ with $n = N \cdot |LD_T|$.

## 4. A Transformer-based Universal Algorithm-Implicit Learner

Drawing on prior work and our own empirical findings, we argue that a promising direction to achieve practical universality lies in *algorithm-implicit* meta-learners that can exploit large-scale meta-datasets and rely on minimal inductive bias. Guided by our theoretical framework, we therefore design our meta-learner as a demonstration-conditioned inference function given by a parametrized black-box function $g_\theta$, which is implemented using a non-causal transformer that processes support and query examples jointly to directly produce predictions for the query. This approach does not require test-time training and makes predictions using only a single forward pass, allowing for efficient deployment under computational constraints. We call our method Transformer-based Algorithm-Implicit Learner (TAIL).

In contrast to previous transformer-based algorithm-implicit approaches, we introduce three technical contributions: **(i)** universal feature encoding via random projections enabling arbitrary $\mathcal{X}_T$; **(ii)** random injection label embeddings, enabling handling arbitrary $\mathcal{Y}_T$ including extrapolation to more classes than seen in training; **(iii)** improved computational efficiency at scale through in-line processing of multiple query samples. With these innovations, our method can be meta-trained on a much broader distribution of tasks (across domains, modalities, and label configurations), which in turn allows learning a more general and powerful implicit algorithm.

Each element of the input sequence to the transformer represents a sample from the support set, including its label, or an unlabeled query sample. For a support set $S = \{(x_i, y_i)\}_{i=1}^n$ with $n := |S| = N \cdot K$ and a query sample $x'$ from $Q$, the input sequence is given by $Z = (z_1, \ldots, z_n, z')$. In practice, we process all query samples together in one sequence for better training efficiency. This speeds up training and testing by orders of magnitude and we experimentally show the computational advantages. For simplicity we will continue to use the former notation with a single query sample. The transformer encoder $\Upsilon$ acts on $Z$ and produces an output sequence $\Upsilon(Z)$.

To handle tasks that differ in data distributions, feature spaces, and label spaces, we employ components hereafter described that (i) map inputs into a shared format for constructing the sequence $Z$, and (ii) project transformer outputs back into the original label space (see Figure 1).

### 4.1. Universal Feature Encoding

To handle varying feature domains $\mathcal{X}_T \neq \mathcal{X}_{T'}$ across tasks we encode the features from $\mathcal{X}_T$ into $\mathbb{R}^{d_T}$ with an encoder $\phi_T$ appropriate for the respective feature modality of each meta-training task. The encoder may take different forms, for example, it could be a simple concatenation of the input features or a pretrained feature extractor. We use pretrained feature extractors for which we provide details in Appendix D. The encoded vector is then projected into $\mathbb{R}^{d_{\text{data}}}$ with a randomly sampled projection $\pi : \mathbb{R}^{d_T} \to \mathbb{R}^{d_{\text{data}}}$. We sample $\pi$ uniformly from extended permutations (see Appendix A.4). This choice gives us two desirable properties. First, it avoids overfitting to the feature structure of a specific encoder $\phi_T$. Second, the random permutation of the feature space essentially acts as data augmentation, enabling our model to see more diverse inputs.

This is similar to how the human brain processes multimodal information. The brain employs modality-specific processing in early sensory cortices that handle low-level features (Calvert, 2001). These specialized regions then feed into convergence zones, where information from different modalities is processed into increasingly abstract representations (Damasio, 1989). This mirrors our approach of using modality-specific encoders followed by a transformer which is shared across domains and reasons over the latent representations.

### 4.2. Random Injection Label Embedding and Classification Head

We consider a general meta-learning setting in which label spaces may differ across tasks, i.e. $\mathcal{Y}_T \not\cong \mathcal{Y}_{T'}$, and where the number of labels at test time may exceed those observed during training. We refer to this setting as *label space extrapolation*. Because attention-based architectures do not require architectural changes for longer sequences, transformers naturally scale to tasks with varying numbers of labels or examples per label. We only need to design a label embedder and a classification head to handle arbitrary $\mathcal{Y}_T$. To this end, we define a global learnable dictionary $\mathcal{E}$ of label embeddings $\{e_1, \ldots, e_M\} \subset \mathbb{R}^{d_{\text{label}}}$ with $M \gg 1$ and define $\mathcal{E}(i) = e_i$ for any index $i \in [M] = \{1, \ldots, M\}$. For any task for which $K := |\mathcal{Y}_T| \leq M$, we can simply use $K$ embeddings from the dictionary, thereby unifying the label space for all tasks. Crucially, if we make $M$ large enough, we can use this dictionary of embeddings for test tasks that have much larger labels sets than any of the training tasks.

To ensure that each of the embeddings is trained, we select the $K$ embeddings to use at random. For each episode with task $T$ we sample an injective mapping $\rho : \mathcal{Y}_T \to [M]$ uniformly from the set of all injections $\text{Inj}(\mathcal{Y}_T, [M])$. The label embedder for this episode is now given by $\mathcal{E} \circ \rho$, which maps elements of $\mathcal{Y}_T$ to vectors in the continuous space $\mathbb{R}^{d_{\text{label}}}$. We prove that this strategy meaningfully trains all embeddings, even when $K \ll M$ for all tasks $T \in \mathcal{T}$, and leads to label space extrapolation ability in Appendix A.

Lastly, we need a classifier head $\Psi$ acting on the transformer output $\Upsilon(Z)$. We use a linear layer $s$ to produce class scores for each index of the label embedding dictionary and compute $\hat{\jmath} = \arg\max_{j \in \rho(\mathcal{Y})} s_j$ where $\rho(\mathcal{Y}) \subset [M]$ is the image of the label space $\mathcal{Y}$ under $\rho$, restricting the possible indices to those of active embeddings in this episode. The classifier head is then given by $\Psi(\Lambda) = \rho^{-1}(\hat{\jmath})$ with $\hat{\jmath} = \arg\max_{j \in \rho(\mathcal{Y})}(s_j(\Lambda))$ reversing the index selection in the embedder.

Input tokens are constructed by concatenating the feature encoding and label embedding. For the query token $z'$ we use a learnable query class marker $c \in \mathbb{R}^{d_{\text{label}}}$ in place of the label embedding.

### 4.3. Training procedure

We train TAIL on a large-scale meta-dataset, consisting of ImageNet (Russakovsky et al., 2015), Meta-Album (Ullah et al., 2022) and MedIMeta (Woerner et al., 2025). We sample training episodes by first sampling a task from the meta-training set $\mathcal{T}_{\text{train}}$ and a "number of shots" $N$ for the episode and then sampling support and query sets as described in Section 3.4. The episode loss is given by the empirical risk $R_Q(f_{g_\theta}) = \sum_{(x,y) \in Q} \ell_T(g_\theta(S, x), y))$. Details about the training procedure and chosen hyperparameters are given in Appendix E.

### 4.4. Theoretical properties

As established in Section 3.3, TAIL must be a valid learning algorithm for varying feature domains and label domains in order to satisfy the requirements of practical universality. We propose that the validity of the implicit algorithm learned by TAIL is invariant to both the feature domain and the label domain. This is ensured by the randomly sampled extended permutation and the random injection label embedding. We theoretically show the invariance in the label domain by providing proofs for the coverage and unbiasedness of the embedding dictionary and invariance to the feature domain by proving the coordinate coverage of the randomly sampled $\pi$ as well as the resulting equivariance to feature coordinate permutations in Appendix A. In order to avoid learning unwanted correlations our algorithm should be invariant to the order of demonstrations and on the coding of the labels. In Appendix A we prove that this is indeed the case.

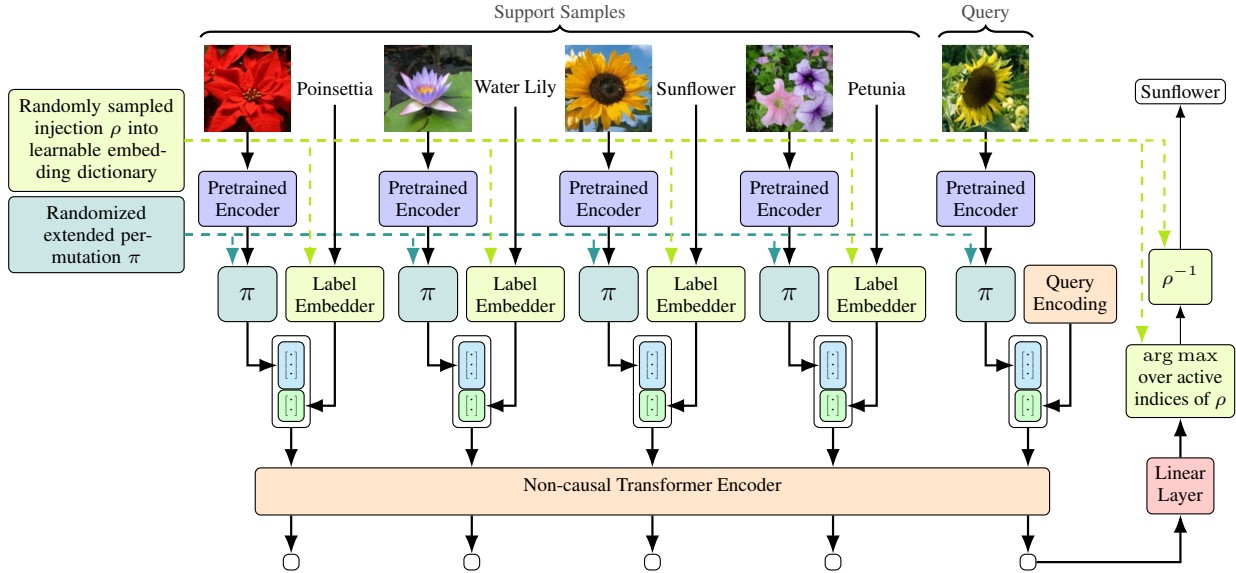

*Figure 1.* **Method overview**. The input is encoded with a modality-appropriate pretrained encoder and then projected to a common modality-agnostic space. The labels are embedded using a randomized injection to a learnable embedding dictionary. The input and label embeddings are concatenated and form the input tokens for a transformer encoder. A linear classification head makes a prediction in label embedding space, which is then remapped to the original set of labels.

## 5. Related Work

Existing meta-learning approaches can be categorized as model-based, optimization-based, or metric-based. Model-based methods such as MANN (Santoro et al., 2016) learn a built-in learning algorithm which adapts to new tasks by changing its internal state. Optimization-based methods, such as Model-Agnostic Meta-Learning (MAML) (Finn et al., 2017), learn an initialization that can be adapted to new tasks with a few gradient updates. Metric-based methods, such as Prototypical Networks (Snell et al., 2017), learn a metric space in which samples from the same class are close. All these approaches either cannot process varying feature and label spaces, or their performance drops substantially when the test domain is significantly different from the meta-training set (Chen et al., 2020; Luo et al., 2023).

It has recently been shown that the naive approach of fine-tuning or linear probing of large pretrained models on a few-shot support set often outperforms meta-learning approaches (Guo et al., 2020; Oh et al., 2022). Moreover, those approaches can be trivially applied to different label spaces by replacing the classification head. However, fine-tuning is computationally expensive when new tasks have to be learned frequently. Furthermore, the weights for each task need to be stored to reuse the model at a later time. A simple alternative to fine-tuning is applying a ProtoNet head to a fixed pretrained backbone which allows meta-testing without any training at test time (see e.g. (Fifty et al., 2023)). However, this approach heavily relies on the quality of the backbone. Prompt-based, adapter-based, or

external-knowledge methods combining multiple foundation models can yield strong performance across domains, but often at the cost of requiring careful prompt design, fine-tuning, or reliance on pre-training domain coverage (Liu et al., 2024). These methods are also typically restricted to a single modality.

Some early model-based meta-learning approaches (Santoro et al., 2016; Kaiser et al., 2017) reformulate few-shot classification as a sequence modeling problem using LSTMs (Hochreiter & Schmidhuber, 1997). More recent approaches pursue a similar strategy using self-attention sequence models (Vaswani et al., 2017). Mishra et al. (2018) apply temporal convolutions alternating with causal attention to the concatenated support and query data. Their model (SNAIL) treats the support set as a sequence of concatenated (input, label) pairs and directly produces query predictions. Due to its architecture requiring a fixed size for the features and labels, SNAIL cannot generalize across modalities or label sets. Newer works use transformers as few-shot learners, drawing parallels to in-context learning in NLP (Brown et al., 2020). GPICL (Kirsch et al., 2024) pursues a similar strategy using a causal transformer model, however, due to the use of positional encodings, the method can only process fixed-size sequences and can therefore not perform label space extrapolation. Because it uses random projections of the input for data augmentation the architecture is, in theory, suitable for processing different modalities.

The causal nature of the above approaches breaks the equivariance to label re-indexing and the demonstration order

*Table 2.* Mean classification accuracy in % over 1000 test episodes.

| Shot | Method | CIFAR-FS | *mini*ImageNet | *tiered*ImageNet | Pascal VOC |
|---|---|---|---|---|---|
| 5-shot | Linear Probe | 91.86 ± 0.32 | 97.65 ± 0.15 | 95.30 ± 0.30 | 83.57 ± 0.53 |
| | ProtoHead | 91.09 ± 0.34 | 97.58 ± 0.15 | 95.54 ± 0.27 | 84.26 ± 0.51 |
| | SNAIL | 91.03 ± 0.38 | 98.93 ± 0.11 | 97.43 ± 0.23 | 85.47 ± 0.53 |
| | GPICL | 91.20 ± 0.35 | 99.44 ± 0.07 | 98.18 ± 0.19 | 87.46 ± 0.51 |
| | CAML | 91.69 ± 0.34 | 99.29 ± 0.08 | 97.98 ± 0.21 | 87.87 ± 0.51 |
| | **TAIL (ours)** | **94.55 ± 0.29** | **99.63 ± 0.06** | **98.67 ± 0.16** | **89.78 ± 0.48** |
| 1-shot | Linear Probe | 75.95 ± 0.64 | 88.22 ± 0.49 | 85.04 ± 0.61 | 64.22 ± 0.68 |
| | ProtoHead | 73.24 ± 0.68 | 87.84 ± 0.51 | 83.76 ± 0.63 | 62.84 ± 0.77 |
| | SNAIL | 76.37 ± 0.65 | 95.79 ± 0.30 | 92.35 ± 0.48 | 72.36 ± 0.75 |
| | GPICL | 77.54 ± 0.66 | 97.64 ± 0.24 | 94.80 ± 0.41 | 75.61 ± 0.70 |
| | CAML | 77.93 ± 0.65 | 97.08 ± 0.24 | 93.66 ± 0.42 | 76.76 ± 0.72 |
| | **TAIL (ours)** | **84.35 ± 0.58** | **98.79 ± 0.14** | **96.76 ± 0.30** | **80.48 ± 0.73** |

invariance (see Section 4.4). CAML by Fifty et al. (2023) instead uses a non-causal transformer and a Equal Length and Maximally Equiangular Set (ELMES) of vectors for embedding labels. Although ELMES vectors are fixed, the transformer needs to be trained with $K$-way tasks to handle $K$-way tasks at test time and therefore CAML can also not perform label space extrapolation. Moreover, it cannot work in the cross-modality setting due to its fixed token size.

Recent work compares in-context learning with meta-learning and studies how transformers learn in context. Oswald et al. (2023) show that transformers can approximate gradient descent in their forward pass. Bai et al. (2023) prove that transformers can implement a range of classical algorithms and perform in-context algorithm selection. Wu et al. (2025) view ICL models as meta-learners, arguing that transformers learn data-dependent learning algorithms. Our work complements this line of research by providing a formal definition of practical universality, introducing the algorithm-explicit/implicit distinction, and by designing a practical algorithm-implicit meta-learner that satisfies these formal properties while achieving state-of-the-art performance. Multiple surveys (Hospedales et al., 2021; Vettoruzzo et al., 2024) present theoretical frameworks for meta-learning.

# 6. Experiments

We evaluate TAIL across four different settings: performance on tasks with a similar domain (i.e. in-domain), cross-domain performance, generalization to unseen modalities, and label space extrapolation. For each test task and each $N$ we sample 1000 episodes with different support and query sets as described in Section 3.4. We report the mean accuracy over these 1000 episodes and the 95%-CI for the mean. Moreover, we assess the computational efficiency compared to the baselines. Our experiments show that TAIL achieves state-of-the-art results while having the flexibility to handle completely new domains and task configurations without retraining. We additionally report ablation studies in Appendix C to analyze the effect of the universal feature encoding using random projections, and the embedding schedule of the embedding dictionary.

## 6.1. Baselines and Training

We consider two algorithm-explicit, and three algorithm-implicit baselines. First, we consider the algorithm-explicit **Linear Probing**, which trains a linear classifier on representations from pretrained foundation models at meta-test time. This can be considered a universal learning algorithm and has been shown to perform well on specialized datasets (Woerner & Baumgartner, 2024), but requires retraining at test time for every task. Next, we consider the algorithm-explicit ProtoNet (Snell et al., 2017) with a fixed pre-trained backbone (which we coin **ProtoHead**), which is not meta-trained, but allows meta-testing without retraining at test time. Lastly, we consider the attention-based algorithm-implicit approaches **SNAIL** (Mishra et al., 2018), **CAML** (Fifty et al., 2023) and **GPICL** (Kirsch et al., 2024). We make slight modifications to GPICL, to able to process different label spaces (see Appendix D.3). For all baseline methods, as well as TAIL (ours), we use fixed pretrained backbones as feature encoders. The encoders are frozen for all methods and are not fine-tuned. To ensure a fair comparison, all meta-learning algorithms were trained on *the same* meta-training data and used *the same* pretrained encoders. We show in Appendix B.2 that TAIL's performance advantage does not depend on a specific encoder.

## 6.2. Results

**In-Domain Image Classification:** In order to evaluate the in-domain performance on standard benchmark problems, we tested all approaches on MiniImageNet (Vinyals et al., 2016), tieredImageNet (Ren et al., 2018), CIFAR-FS (Bertinetto et al., 2018), and Pascal VOC (Everingham et al., 2010), which are generic object-recognition datasets and therefore can be considered in-domain with respect to our training set. As shown in Table 2, TAIL consistently outperformed all baselines in the 1-shot and 5-shot setting.

**Cross-Domain Specialized Datasets:** For the cross-domain evaluation, we tested all learners on a diverse set of specialized domains not present in our meta-training set: Caltech Birds Dataset (CUB) (Wah et al., 2011), FGVC-Aircraft (Maji et al., 2013), meta-iNat and tiered meta-iNat (Wertheimer & Hariharan, 2019), the cxr, oct and pbc subsets from MedIMeta (Woerner et al., 2025), the Paintings dataset (Crowley & Zisserman, 2015) and Inter-Domain Image Classification Pascal+Paintings (Fifty et al., 2023).

The 5-shot and 1-shot performance in the out-of-domain setting are shown in Table 3. Some datasets may contain certain classes with a semantic overlap with the training set. These are marked with an asterisk. The medical datasets from MedIMeta were curated to exclude classes with semantic overlap. TAIL achieved state-of-the-art performance on the majority of datasets and is competitive on the two remaining datasets, without any domain-specific retraining.

*Table 3.* Mean 5-way classification accuracy in % over 1000 test episodes for 1-shot and 5-shot settings.

| | | Aircraft* | CUB* | meta-iNat* | tiered meta-iNat* | cxr | oct | pbc | Paintings* | Pascal-Paintings* |
|---|---|---|---|---|---|---|---|---|---|---|
| **5-shot** | **Linear Probe** | 92.12 ± 0.42 | 95.17 ± 0.30 | 92.91 ± 0.36 | 88.32 ± 0.47 | **25.10 ± 0.40** | 49.61 ± 0.55 | 68.39 ± 0.54 | **66.21 ± 0.44** | 71.62 ± 0.43 |
| | **ProtoHead** | 92.07 ± 0.42 | 95.26 ± 0.30 | 92.51 ± 0.39 | 88.47 ± 0.44 | 25.04 ± 0.41 | 49.37 ± 0.53 | 69.81 ± 0.51 | 66.06 ± 0.46 | 71.57 ± 0.44 |
| | **SNAIL** | 90.19 ± 0.48 | 95.57 ± 0.34 | 95.08 ± 0.33 | 91.21 ± 0.47 | 21.52 ± 0.30 | 35.36 ± 0.46 | 71.08 ± 0.54 | 57.49 ± 0.46 | 69.12 ± 0.46 |
| | **GPICL** | 90.83 ± 0.45 | 96.02 ± 0.30 | 95.56 ± 0.31 | 91.15 ± 0.46 | 22.70 ± 0.34 | 41.64 ± 0.50 | 53.28 ± 0.52 | 54.29 ± 0.48 | 65.79 ± 0.51 |
| | **CAML** | 93.09 ± 0.39 | 97.55 ± 0.22 | 96.20 ± 0.28 | 93.53 ± 0.36 | 23.30 ± 0.40 | 46.48 ± 0.53 | 80.19 ± 0.45 | 58.72 ± 0.49 | 70.13 ± 0.46 |
| | **TAIL (ours)** | **95.01 ± 0.34** | **98.51 ± 0.17** | **97.70 ± 0.21** | **95.69 ± 0.31** | 23.68 ± 0.38 | **50.64 ± 0.52** | **84.96 ± 0.42** | 63.03 ± 0.48 | **73.04 ± 0.47** |
| **1-shot** | **Linear Probe** | 80.32 ± 0.70 | 83.06 ± 0.66 | 79.02 ± 0.68 | 71.91 ± 0.72 | **22.35 ± 0.35** | 36.29 ± 0.53 | 45.91 ± 0.59 | **49.96 ± 0.56** | 51.14 ± 0.56 |
| | **ProtoHead** | 78.89 ± 0.72 | 81.95 ± 0.71 | 78.31 ± 0.69 | 70.48 ± 0.75 | 22.33 ± 0.36 | 36.25 ± 0.52 | 45.29 ± 0.58 | 48.64 ± 0.58 | 50.58 ± 0.54 |
| | **SNAIL** | 80.82 ± 0.70 | 88.96 ± 0.58 | 87.30 ± 0.60 | 81.81 ± 0.68 | 20.61 ± 0.30 | 29.93 ± 0.45 | 60.26 ± 0.67 | 45.01 ± 0.57 | 51.82 ± 0.61 |
| | **GPICL** | 74.30 ± 0.80 | 85.63 ± 0.65 | 88.92 ± 0.57 | 78.29 ± 0.74 | 20.72 ± 0.29 | 30.71 ± 0.44 | 30.63 ± 0.48 | 41.63 ± 0.56 | 49.91 ± 0.57 |
| | **CAML** | 84.33 ± 0.65 | 92.34 ± 0.49 | 90.74 ± 0.52 | 84.28 ± 0.64 | 21.86 ± 0.36 | 35.53 ± 0.54 | 61.73 ± 0.67 | 45.77 ± 0.60 | 53.28 ± 0.63 |
| | **TAIL (ours)** | **89.42 ± 0.56** | **95.51 ± 0.38** | **93.84 ± 0.42** | **90.23 ± 0.53** | 22.15 ± 0.38 | **36.74 ± 0.58** | **70.25 ± 0.66** | 48.00 ± 0.61 | **55.71 ± 0.63** |

*Table 4.* Cross-modal performance: models were trained on image classification tasks and tested on text and audio classification tasks.

| | Text (IMDB) | | Audio (Music Genre) | |
|---|---|---|---|---|
| | 5-shot | 1-shot | 5-shot | 1-shot |
| Linear Probe | 89.33 ± 0.44 | **85.32 ± 1.03** | 38.02 ± 0.57 | 29.58 ± 0.51 |
| ProtoHead | 88.92 ± 0.44 | 83.86 ± 1.13 | 54.74 ± 0.58 | 39.09 ± 0.63 |
| GPICL (trained on images) | 50.88 ± 0.42 | 50.31 ± 0.28 | 20.03 ± 0.08 | 19.98 ± 0.13 |
| TAIL (ours) (trained on images) | **89.62 ± 0.48** | 84.87 ± 1.04 | **55.33 ± 0.60** | **40.63 ± 0.62** |

**Cross-Modal Generalization to Unseen Modalities:** The key test of practical universality is the ability to generalize to completely different modalities without retraining. To assess the limits of generalizability, we evaluated the models trained exclusively on images on a text classification and an audio classification problem. Here, we only included the baselines which architecturally permit operating in a different feature space than was used in the meta-training stage: Linear Probing, ProtoHead, GPICL and TAIL. We used sentiment classification of IMDB movie reviews (Maas et al., 2011) and music genre classification on the GTZAN dataset (Tzanetakis & Cook, 2002) as the evaluation tasks.

The results in Table 4 show that TAIL achieves superior cross-modal generalization on both text and audio. On text classification, TAIL outperforms all approaches in the 5-shot setting and is only slightly outperformed by Linear Probing in the 1-shot setting. On audio classification, TAIL outperforms all baselines in both settings. Out of the meta-learned methods, TAIL maintains the strongest performance when applied to completely different modalities. While GPICL can theoretically handle different modalities, its performance degrades severely, to the point that the accuracy is on the level of random chance.

**Label Space Extrapolation:** Traditional meta-learning methods fail when confronted with tasks containing more classes than seen during training. To illustrate that TAIL gracefully handles label space extrapolation, we used the meta-trained learners from Section 6.2, which were trained only on task instances with $K \leq 5$. We then evaluated performance with increasing numbers of classes up to 100-way classification. We also took advantage of TAIL's computational efficiency to train a version with 50 labels used in the meta-training stage (TAIL 50w), which is computationally infeasible for the other attention-based approaches.

As can be seen in Figure 2 (left), performance degraded with more labels $K$ per tasks as is expected. TAIL achieved the top performance until 70-way classification tasks, where it was outperformed by Linear Probing and ProtoHead which require a domain-specific classification head. We further note that TAIL trained with tasks up to 50 labels significantly outperforms the baselines throughout the testing scenario.

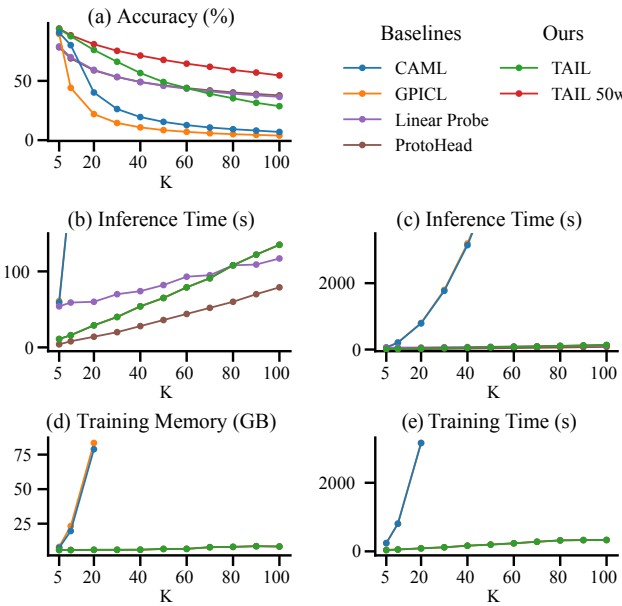

*Figure 2.* **(a)**: performance degradation with increasing number of classes (1-shot setting). **(b)** and **(c)**: wall clock time for 1000 test episodes as a function of task size. Two different scales show the relation to the algorithm-explicit baselines and to the meta-learning baselines. **(d)**: memory usage during training as a function of task size, **(e)**: wall clock time for 1000 training episodes.

**Computational Efficiency for Large Label Sets:** While GPICL (Kirsch et al., 2024) and CAML (Fifty et al., 2023) can theoretically handle arbitrary label spaces, in practice they are severely computationally limited at larger task sizes. While evaluation with large label spaces is still possible up to a point, training with label spaces larger than 20 is computationally prohibitive on current computational infrastructures. In contrast, TAIL provides dramatic computational advantages over existing attention-based meta-learning ap-

*Table 5.* Average accuracy (%) with ResNet-18 encoder (encoder swapped test time only).

|  | In-domain | | Cross-domain | |
| --- | --- | --- | --- | --- |
|  | 5-shot | 1-shot | 5-shot | 1-shot |
| Linear Probe | 76.25 ± 0.26 | 57.07 ± 0.34 | 54.20 ± 0.18 | 40.33 ± 0.19 |
| ProtoHead | 87.59 ± 0.21 | 70.58 ± 0.33 | 63.67 ± 0.16 | 46.49 ± 0.20 |
| TAIL (ours) | **88.91 ± 0.21** | **79.46 ± 0.31** | **66.08 ± 0.16** | **53.32 ± 0.20** |

proaches. To illustrate this, we measured the wall clock time for solving a 1-shot meta-test task with increasing numbers of labels ($K$). As can be seen in Figure 2 (c), the wall clock time of GPICL and CAML increases very rapidly with increasing $K$, while TAIL retains a similar computational complexity to the methods not based on transformers (i.e. Linear Probing and ProtoHead). Training time and memory usage (Figure 2 (d,e)) show an even more dramatic difference. Training GPICL and CAML becomes infeasible for $K \geq 20$. Figure 2 (b) shows that TAIL is in fact faster than Linear Probing for typical task sizes below $K = 70$ since it does not require training at meta-test time. The results for the 5-way task look similar (see Appendix B.1). We measured only the time and memory requirements of TAIL and the baselines without the pretrained encoder.

**Generalization to Different Encoders:** To isolate the contribution of the meta-learning algorithm from the pretrained encoder, we replaced the ViT-H encoder with a substantially weaker ResNet-18 (pretrained on ImageNet-1k) at test time only. TAIL was still meta-trained with ViT-H features. This tests how well the meta-learned algorithm transfers when the underlying representations change drastically. Table 5 shows the average accuracy across in-domain and cross-domain datasets (full per-dataset results in Appendix B.2).

With the weaker encoder, all methods drop considerably, but TAIL consistently outperforms the baselines. Notably, TAIL's advantage is *larger* with ResNet-18 than with ViT-H, demonstrating that the meta-learned algorithm contributes more when the encoder is weaker.

## 7. Discussion and Conclusion

We introduced a theoretical framework for meta-learning that addresses the lack of precise definitions in prior work. Our framework provides two main contributions: the notion of *practical universality*, which formally specifies what it means for a learning algorithm to generalize across varying feature domains, label spaces, and data distributions with finite-sample guarantees, and a taxonomy distinguishing between *algorithm-explicit vs. algorithm-implicit* approaches, which explains why certain meta-learning paradigms generalize better than others and provides a principled lens for future work.

Guided by this framework, we developed TAIL, a meta-learning approach designed specifically to satisfy practical

universality and improving upon prior work. Our technical contributions include random projections for cross-modal generalization and label embeddings with a global dictionary to generalize to new label domains, including domains with far more classes than seen in training.

Empirically, TAIL sets new state-of-the-art results, handles tasks with label spaces far larger than those seen during training, with up to $20\times$ more classes, and offers large computational savings. Most crucially, it transfers across unseen modalities without retraining and achieves strong performance on text and audio classification after training only on images.

Although TAIL is less reliant on encoder quality than encoder-only baselines (as shown in Section 6.2), it still benefits from strong pretrained representations for state-of-the-art performance. The practical universality of TAIL is currently limited to classification. While the theoretical framework is task-type-agnostic, extending TAIL to other task types remains future work. Future research directions include extending algorithm-implicit learning to regression and structured prediction, and investigating whether similar advantages arise in reinforcement learning.

## Acknowledgements

Funded by the Deutsche Forschungsgemeinschaft (DFG, German Research Foundation) under Germany's Excellence Strategy – EXC number 2064/1 – Project number 390727645. The authors thank the International Max Planck Research School for Intelligent Systems (IMPRS-IS) for supporting Stefano Woerner.

## Impact Statement

From a positive societal perspective, practically universal few-shot learners may contribute to more data-efficient and computationally efficient AI systems. This could reduce the environmental footprint associated with repeated retraining and fine-tuning, and enable broader access to machine learning technologies beyond organizations with large-scale data and compute resources. The ability to transfer across modalities without architectural changes may also support faster prototyping and reuse of models across disciplines.

At the same time, increased generality introduces risks if such systems are deployed without sufficient domain understanding or validation. A model that adapts broadly across tasks may be applied in inappropriate contexts or trusted beyond its empirical guarantees, particularly in high-stakes domains. While this work focuses on methodological foundations and controlled benchmarks, responsible deployment would require domain-specific evaluation, uncertainty awareness, and human oversight.

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

# A. Theoretical Analysis and Proofs

In this Appendix, we formalize and prove the theoretical properties described in Section 4 of the main article.

## A.1. Permutation Equivariance with respect to the Label Space

**Theorem A.1** (Equivariance to label re-indexing). *Let $\mathcal{X}$ be a feature space, $\mathcal{Y}$ a label space and $S = \{(x_i, y_i)\}_{i=1}^n$ a support dataset and $(x, y)$ a query sample. For any permutation $\sigma$ of $\mathcal{Y}$, let*

$$S^\sigma = \{(x_i, \sigma(y_i))\}_i, \quad y_q^\sigma = \sigma(y_q).$$

*Then*

$$g(S^\sigma, x) \overset{d}{=} \sigma(g(S, x)).$$

*i.e. $g_\theta(S, x)$ is equivariant in distribution to the reindexing of $\mathcal{Y}$.*

*Proof of Theorem A.1.* Let $\mathcal{E}$ be our embedding dictionary and $\rho : \mathcal{Y} \to [M]$ be an injection sampled uniformly from the set of all injections $\mathrm{Inj}(\mathcal{Y}_T, [M])$. Define $\rho' := \rho \circ \sigma^{-1}$. We note that the images under $\rho$ and $\rho'$ are equal, i.e. $\rho(\mathcal{Y}) = \rho'(\mathcal{Y})$. Then

$$g(S^\sigma, x\,;\rho') = \rho'^{-1}\left(\arg\max_{j \in \rho'(\mathcal{Y})} s_j\left(\Upsilon\left(\begin{bmatrix} \pi(\phi(x_1)) & \dots & \pi(\phi(x_n)) & \pi(\phi(x)) \\ \mathcal{E}(\rho'(\sigma(y_1))) & \dots & \mathcal{E}(\rho'(\sigma(y_n))) & c \end{bmatrix}\right)\right)\right)$$

$$= \sigma\left(\rho^{-1}\left(\arg\max_{j \in \rho(\mathcal{Y})} s_j\left(\Upsilon\left(\begin{bmatrix} \pi(\phi(x_1)) & \dots & \pi(\phi(x_n)) & \pi(\phi(x)) \\ \mathcal{E}(\rho(y_1)) & \dots & \mathcal{E}(\rho(y_n)) & c \end{bmatrix}\right)\right)\right)\right) = \sigma\left(g(S, x\,;\rho)\right).$$

Since $\rho$ is sampled uniformly over injections, $\rho' := \rho \circ \sigma^{-1}$ and $\rho$ have the same probability. Combining, the prediction distributions are identical. $\square$

## A.2. Coverage of the Embedding Dictionary

Even though each episode only uses $K$ out of $M$ embeddings, we show that across episodes all embeddings and their corresponding "detectors" in the transformer are trained.

**Proposition A.2** (Unbiased gradients). *For each embedding $e_j$, the episode gradient satisfies*

$$\mathbb{E}_\rho[\nabla_{e_j}\ell] = \tfrac{K}{M}\,\mathbb{E}\big[\nabla_{e_j}\ell \mid j \in S\big].$$

*Thus stochastic gradients are unbiased up to the constant factor $\frac{K}{M}$.*

**Proposition A.3** (Coverage over $t$ episodes). *Let $N_j(t)$ be the number of episodes in which $e_j$ is included. Then*

$$N_j(t) \sim \mathrm{Binomial}\big(t, \tfrac{K}{M}\big), \quad \mathbb{E}[N_j(t)] = \tfrac{tK}{M}.$$

*By Chernoff bound, for any $\delta \in (0, 1)$,*

$$\Pr\left[N_j(t) \le (1-\delta)\tfrac{tK}{M}\right] \le \exp\left(-\frac{\delta^2 tK}{2M}\right).$$

*It follows that with high probability every embedding is updated $\Omega(\frac{tK}{M})$ times once $t \gtrsim \frac{M}{K}\log M$.*

*Remark* A.4. Since every episode includes at least one label embedding, transformer parameters $\theta_\Upsilon$ interacting with embeddings receive gradient updates in every episode. By symmetry of $\rho$, these "detectors" are trained uniformly across all embeddings.

## A.3. Demonstration Order Invariance

It is a desirable quality for the DCI predictor to be invariant to the order in which the support set is presented. In the following theorem we show that this is indeed the case for TAIL.

**Theorem A.5** (Demonstration Order Invariance). *Let $S$ be a sequence of support samples $S = ((x_i, y_i))_{i=1}^n$ and $(x, y)$ a query sample. For any permutation $\sigma$ of $S$*

$$g(\sigma(S), x) = g(S, x).$$

*i.e. $g_\theta(S, x)$ is invariant to the order of demonstrations.*

*Proof.* Building on (Kossen et al., 2021, Appendix A), we only need to prove that the domain-specific encoder and the random injection embedding are equivariant to the order of demonstrations. Since $\phi, \rho$ are elementwise applied the input sequence and do not depend on position, it trivially follows that embedding the sequence is permutation equivariant. Since the transformer $\Upsilon$ itself is permutation invariant following (Kossen et al., 2021, Appendix A), and $\Psi$ only operates on the index of the query sample, $g_\theta$ is invariant to permutations. $\qquad\square$

### A.4. Extended Permutations

**Definition A.6** (Extended permutation). Let $n \leq k$. An *extended permutation matrix* is a binary matrix $E \in \{0, 1\}^{k \times n}$ such that each column contains exactly one 1, each row contains at most one 1. Equivalently, $E$ encodes an injective map $\pi : \{1, \ldots, n\} \to \{1, \ldots, k\}$.

### A.5. Feature Domain Invariance

We now prove that the validity of TAIL as a learning algorithm is invariant to the feature domain $\mathcal{X}_T$. The key mechanism enabling this invariance is the random extended permutation $\pi$ that projects encoded features into the common latent space.

**Definition A.7** (Coordinate-Symmetric Learning). A DCI function $g_\theta$ exhibits coordinate-symmetric learning if for any permutation $\sigma$ of the coordinates of the latent space $\mathbb{R}^{d_{\text{data}}}$, the function $g_\theta$ trained with projections $\{\pi_i\}$ has the same expected performance as $g_\theta$ trained with projections $\{\sigma \circ \pi_i\}$.

**Proposition A.8** (Coordinate Coverage). *Let $\pi : \mathbb{R}^{d_T} \to \mathbb{R}^{d_{\text{data}}}$ be an extended permutation sampled uniformly from $\mathrm{Inj}([d_T], [d_{\text{data}}])$. Let $N_j(t)$ be the number of training episodes in which coordinate $j \in [d_{\text{data}}]$ receives a non-zero input. Then:*

$$N_j(t) \sim \mathrm{Binomial}\left(t, \frac{d_T}{d_{\text{data}}}\right), \quad \mathbb{E}[N_j(t)] = \frac{t \cdot d_T}{d_{\text{data}}}.$$

*By Chernoff bound, for any $\delta \in (0, 1)$:*

$$\Pr\left[N_j(t) \leq (1 - \delta)\frac{t \cdot d_T}{d_{\text{data}}}\right] \leq \exp\left(-\frac{\delta^2 t \cdot d_T}{2 \cdot d_{\text{data}}}\right).$$

*Thus, with high probability, every coordinate of the latent space is updated $\Omega\left(\frac{t \cdot d_T}{d_{\text{data}}}\right)$ times once $t \gtrsim \frac{d_{\text{data}}}{d_T} \log d_{\text{data}}$.*

**Theorem A.9** (Equivariance to Feature Coordinate Permutation). *Let $\phi_T : \mathcal{X}_T \to \mathbb{R}^{d_T}$ be an encoder and $\sigma$ be any permutation of $[d_T]$. Define the permuted encoder $\phi_T^\sigma(x) = \sigma(\phi_T(x))$. Then for any support set $S$ and query $x$:*

$$g_\theta(S, x; \pi \circ \phi_T) \stackrel{d}{=} g_\theta(S, x; \pi \circ \phi_T^\sigma)$$

*where the equality is in distribution over the random choice of $\pi$.*

*Proof.* Let $\pi$ be sampled uniformly from $\mathrm{Inj}([d_T], [d_{\text{data}}])$. Define $\pi' = \pi \circ \sigma^{-1}$. Since $\sigma$ is a bijection on $[d_T]$, we have that $\pi'$ is also an injection from $[d_T]$ to $[d_{\text{data}}]$, and moreover $\pi'$ is uniformly distributed over $\mathrm{Inj}([d_T], [d_{\text{data}}])$.

Now observe:

$$\pi(\phi_T^\sigma(x)) = \pi(\sigma(\phi_T(x))) = (\pi \circ \sigma)(\phi_T(x)) = \pi'(\phi_T(x)) \circ \sigma \circ \sigma^{-1} = \pi'(\phi_T(x)).$$

Since $\pi$ and $\pi \circ \sigma$ have the same distribution (uniform over injections), the projected features $\pi(\phi_T^\sigma(x))$ and $\pi(\phi_T(x))$ have the same distribution. Therefore, $g_\theta(S, x; \pi \circ \phi_T^\sigma) \stackrel{d}{=} g_\theta(S, x; \pi \circ \phi_T)$. $\qquad\square$

**Corollary A.10** (Feature Domain Invariance). *Let $T = (\mathcal{X}_T, \mathcal{Y}_T, p_T, \ell_T)$ and $T' = (\mathcal{X}_{T'}, \mathcal{Y}_{T'}, p_{T'}, \ell_{T'})$ be two tasks with potentially different feature domains. Let $\phi_T : \mathcal{X}_T \to \mathbb{R}^{d_T}$ and $\phi_{T'} : \mathcal{X}_{T'} \to \mathbb{R}^{d_{T'}}$ be encoders such that the encoded features preserve task-relevant structure (i.e., samples from the same class have similar encoded representations). Then the expected risk of $g_\theta$ depends only on the quality of the encoder's representations, not on the specific coordinate structure of the encoder output or the original feature domain $\mathcal{X}_T$.*

*Proof.* By Theorem A.9, the DCI function is equivariant in distribution to any permutation of the encoder's output coordinates. By Proposition A.8, the transformer is trained symmetrically with respect to all coordinates of the latent space. Therefore, the transformer learns a function that depends only on the relationships between projected features (e.g., distances, angles), not on their absolute coordinate positions. Since the extended permutation preserves pairwise relationships up to coordinate relabeling, and the transformer treats all coordinates equivalently, the learned algorithm generalizes to any encoder whose output preserves task-relevant structure. □

## B. Additional Results

### B.1. 5-Shot Accuracy Degradation

We additionally report the accuracy degradation with increasing $K$ on 5-shot tasks in Figure 3.

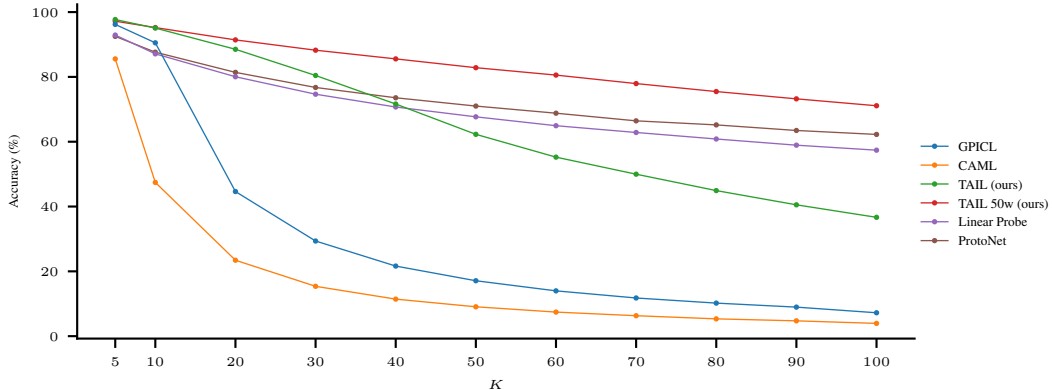

*Figure 3.* Performance degradation with increasing number of classes (5-shot setting).

### B.2. Per-Dataset Results with ResNet-18 Encoder

Tables 6 and 7 show the full per-dataset results with a Resnet-18 encoder at test time.

*Table 6.* In-domain results with ResNet-18 encoder (mean accuracy % over 1000 episodes).

| Shot | Method | CIFAR-FS | *mini*ImageNet | *tiered*ImageNet | Pascal VOC |
|------|--------|----------|----------------|------------------|------------|
| **5-shot** | **Linear Probe** | $64.70 \pm 0.61$ | $88.69 \pm 0.37$ | $85.62 \pm 0.46$ | $66.01 \pm 0.63$ |
| | **ProtoHead** | $77.64 \pm 0.55$ | $97.68 \pm 0.16$ | $95.53 \pm 0.28$ | $79.51 \pm 0.57$ |
| | **TAIL (ours)** | $\mathbf{78.93 \pm 0.56}$ | $\mathbf{98.53 \pm 0.12}$ | $\mathbf{96.86 \pm 0.24}$ | $\mathbf{81.30 \pm 0.59}$ |
| **1-shot** | **Linear Probe** | $46.68 \pm 0.65$ | $68.35 \pm 0.69$ | $66.45 \pm 0.72$ | $46.78 \pm 0.65$ |
| | **ProtoHead** | $55.10 \pm 0.70$ | $85.50 \pm 0.58$ | $82.86 \pm 0.63$ | $58.86 \pm 0.74$ |
| | **TAIL (ours)** | $\mathbf{63.17 \pm 0.79}$ | $\mathbf{95.08 \pm 0.31}$ | $\mathbf{91.64 \pm 0.45}$ | $\mathbf{67.96 \pm 0.76}$ |

*Table 7.* Cross-domain results with ResNet-18 encoder (mean 5-way accuracy % over 1000 episodes).

| Shot | Method | Aircraft* | CUB* | meta-iNat* | tiered meta-iNat* | cxr | oct | pbc | Paintings* | Pascal-Paintings* |
|------|--------|-----------|------|------------|-------------------|-----|-----|-----|------------|-------------------|
| **5-shot** | **Lin. Probe** | $54.58 \pm 0.61$ | $76.46 \pm 0.62$ | $77.67 \pm 0.58$ | $71.10 \pm 0.58$ | $23.70 \pm 0.40$ | $41.35 \pm 0.54$ | $41.62 \pm 0.48$ | $47.98 \pm 0.47$ | $53.35 \pm 0.47$ |
| | **ProtoHead** | $67.30 \pm 0.63$ | $89.72 \pm 0.47$ | $88.69 \pm 0.45$ | $84.46 \pm 0.47$ | $25.57 \pm 0.41$ | $46.04 \pm 0.52$ | $46.93 \pm 0.51$ | $57.75 \pm 0.46$ | $66.56 \pm 0.44$ |
| | **TAIL (ours)** | $\mathbf{70.11 \pm 0.63}$ | $\mathbf{92.12 \pm 0.40}$ | $\mathbf{91.61 \pm 0.39}$ | $\mathbf{88.73 \pm 0.43}$ | $\mathbf{26.59 \pm 0.44}$ | $\mathbf{47.60 \pm 0.53}$ | $\mathbf{52.47 \pm 0.53}$ | $\mathbf{58.14 \pm 0.46}$ | $\mathbf{67.36 \pm 0.43}$ |
| **1-shot** | **Lin. Probe** | $38.95 \pm 0.60$ | $56.88 \pm 0.71$ | $58.32 \pm 0.72$ | $51.14 \pm 0.67$ | $21.72 \pm 0.35$ | $32.52 \pm 0.52$ | $30.56 \pm 0.46$ | $35.40 \pm 0.54$ | $37.52 \pm 0.53$ |
| | **ProtoHead** | $44.36 \pm 0.65$ | $66.89 \pm 0.73$ | $70.18 \pm 0.75$ | $62.88 \pm 0.73$ | $22.58 \pm 0.38$ | $33.67 \pm 0.50$ | $32.15 \pm 0.47$ | $40.54 \pm 0.56$ | $45.12 \pm 0.59$ |
| | **TAIL (ours)** | $\mathbf{50.45 \pm 0.68}$ | $\mathbf{81.92 \pm 0.63}$ | $\mathbf{79.70 \pm 0.68}$ | $\mathbf{73.72 \pm 0.75}$ | $\mathbf{23.11 \pm 0.41}$ | $\mathbf{36.16 \pm 0.54}$ | $\mathbf{37.82 \pm 0.54}$ | $\mathbf{45.22 \pm 0.58}$ | $\mathbf{51.76 \pm 0.62}$ |

With the weaker ResNet-18 encoder, the performance of all methods drops considerably compared to the results obtained with ViT-H. However, TAIL consistently and significantly outperforms Linear Probing and ProtoHead across all datasets. Notably, TAIL's advantage over the encoder-only baselines is larger with ResNet-18 than with ViT-H, demonstrating that the meta-learning algorithm contributes more when the encoder is weaker.

## C. Ablation Studies

In order to better understand our method's reliance of our key architectural components, we conducted the following ablation experiments.

**Random Feature Projection:** First we compared TAIL, to a version of TAIL with random projections without the restriction to extended permutations and to a version of TAIL *without* the universal feature encoding using random projections of the feature space described in Section. 4.1. As can be seen in Table 8, TAIL's performance decreased without the random extended permutation, especially in the cross-domain and cross-modality settings. This indicates that the random projection mechanism is necessary for generalization to unseen domains.

*Table 8.* Average performance on in-domain datasets, cross-domain datasets and cross-modality datasets with and without the random permutation into a common latent space.

|  | 5-shot | | | 1-shot | | |
|---|---|---|---|---|---|---|
|  | in-domain | cross-domain | cross-modality | in-domain | cross-domain | cross-modality |
| TAIL *without* random $\pi$ | 98.75 ± 0.06 | 83.07 ± 0.11 | 87.43 ± 0.62 | 95.80 ± 0.16 | 63.94 ± 0.17 | 66.69 ± 1.62 |
| TAIL, random projection $\pi$ | 99.09 ± 0.07 | 86.83 ± 0.10 | 89.91 ± 0.44 | 96.60 ± 0.13 | 67.70 ± 0.16 | 84.96 ± 0.98 |
| **TAIL, random extended permutation $\pi$** | 99.21 ± 0.06 | 87.58 ± 0.10 | 89.62 ± 0.48 | 97.30 ± 0.12 | 67.80 ± 0.17 | 84.87 ± 1.04 |

**Causal vs. Non-Causal Architecture:** To quantify the impact of removing the causal mask, we replace TAIL's non-causal transformer with an architecture identical in all respects except for a standard causal attention mask. Across all settings, the causal variant exhibits a consistent and substantial drop in accuracy.

*Table 9.* Average performance on in-domain datasets, cross-domain datasets and cross-modality datasets with a causal and with a non-causal transformer architecture.

|  | 5-shot | | | 1-shot | | |
|---|---|---|---|---|---|---|
|  | in-domain | cross-domain | cross-modality | in-domain | cross-domain | cross-modality |
| TAIL with causal architecture | 98.81 ± 0.06 | 70.60 ± 0.13 | 70.08 ± 0.71 | 93.67 ± 0.19 | 53.74 ± 0.17 | 66.91 ± 1.42 |
| **TAIL (non causal)** | 99.21 ± 0.06 | 87.58 ± 0.10 | 89.62 ± 0.48 | 97.30 ± 0.12 | 67.80 ± 0.17 | 84.87 ± 1.04 |

**Mixed-Modality Training:** We explored the impact of adding text data to the meta-training set. As Table 10 shows, the mixed-modality training does not significantly improve performance, except for the text classification task itself. We believe this is an artifact of the composition of the meta-dataset: the comparatively small amount of text data does not significantly increase the data diversity provided by our large image-classification meta-training set. We speculate that exposing TAIL to a larger cross-modal meta-training set with a broader variety of feature domains could strengthen the learned algorithm and add robustness on cross-modality evaluation. The very minor improvement on the text-classification task supports the claim the TAIL generalizes well to unseen modalities.

*Table 10.* Average performance on in-domain datasets, cross-domain datasets and cross-modality datasets with our default meta-training set and with a mixed-modality meta-training set including text-classifcation tasks.

|  | 5-shot | | | 1-shot | | |
|---|---|---|---|---|---|---|
|  | in-domain | cross-domain | cross-modality | in-domain | cross-domain | cross-modality |
| TAIL trained on mixed-modality training set | 99.06 ± 0.06 | 85.19 ± 0.09 | 90.10 ± 0.59 | 96.75 ± 0.15 | 68.91 ± 0.20 | 85.91 ± 1.03 |
| **TAIL (trained on images only)** | 99.21 ± 0.06 | 87.58 ± 0.10 | 89.62 ± 0.48 | 97.30 ± 0.12 | 67.80 ± 0.17 | 84.87 ± 1.04 |

**Training Schedule for the Label Embedding Dictionary:** Lastly, we investigated the effect the embedding dictionary schedule (see Section 4.2) on the speed of convergence. Figure 4 shows that slowly adding more embeddings to the embedding dictionary during training accelerates convergence. This is likely due to the fact that training can be jump-started with easier problems, an effect that is also known from curriculum learning (Bengio et al., 2009).

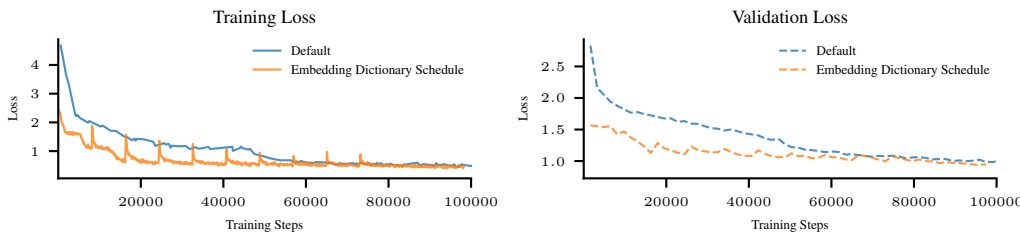

*Figure 4.* Validation loss curves for scheduled addition of more embeddings to the embedding dictionary.

## D. Architecture Details

### D.1. Pretrained Feature Encoders

We use the following pretrained encoders for the different modalities:

**Vision Tasks:** We use the ViT-H model trained on LAION-2B (Schuhmann et al., 2022) provided by OpenCLIP (Ilharco et al., 2021), an open source reimplementation of OpenAI's CLIP. Images were resized to $224 \times 224$ and we applied standard ImageNet normalization.

**Text Tasks:** We use a pretrained uncased version of DistilBERT (Sanh et al., 2020) to embed text tasks.

**Audio Tasks:** We use a pretrained wav2vec 2.0 model (Baevski et al., 2020) to embed audio tasks.

### D.2. Transformer Architecture

We provide the parameters of the transformer architecture of TAIL in Table 11.

*Table 11.* TAIL architecture hyperparameters

| Component | Configuration |
| --- | --- |
| **Transformer Encoder** | |
| Hidden dimension | 1536 |
| Number of layers | 16 |
| Attention heads | 16 |
| MLP dimension | 3072 |
| Dropout | 0.0 |
| Activation | GELU |
| Normalization | Layer Norm |
| **Feature Projection** | |
| Output dimension ($d_{\text{data}}$) | 1280 |
| Type | Extended permutation matrix |
| **Label Embedding** | |
| Embedding dimension ($d_{\text{label}}$) | 256 |
| Dictionary size ($M$) | 100 (default), 256 (large) |

### D.3. Modifications to GPICL

We extend the positional encoding to have more vectors than required and use the first $K \cdot N$ vectors to encode the positions of a sequence. With this approach, GPICL accepts sequences of variable lengths, allowing us to test it in the cross-modality and label extrapolation settings.

# E. Training Details

### E.1. Episode Sampling

For each episode, a dataset is sampled at random from the meta-training set, weighted by the number of classes in the dataset. A task is generated by choosing $K$ classes at random from the dataset. Support and Query sets are then generated by first sampling a "number of shots" $N$ for the episode and then sampling support and query sets as described in Section 3.4.

### E.2. Optimization

We use the Adam optimizer with a circular learning rate schedule and a maximum learning rate of $3 \cdot 10^{-5}$.

