# OpenReview forum: "Universal Algorithm-Implicit Learning"
_ICML.cc/2026/Conference — ICML 2026 regular_

### Official Review · Reviewer_JWbY · 2026-03-11

**Soundness:** 3
**Presentation:** 2
**Significance:** 3
**Originality:** 3
**Overall Recommendation:** 3
**Confidence:** 4

**Summary:**

This paper proposes a theoretical framework for universal meta-learning and introduces TAIL, a transformer-based meta-learner designed to handle tasks with varying domains, modalities, and label spaces. The method uses random projections, randomized label embeddings, and inline query processing to support cross-modal generalization, label-space extrapolation, and efficient inference. Experimental results show strong few-shot performance, improved generalization to unseen domains and modalities, and better scalability than prior transformer-based meta-learning methods.

**Compliance With Llm Reviewing Policy:**

Affirmed.

**Final Justification:**

The rebuttal clarifies several points, but it does not materially change my overall assessment. I still view the method as a relatively incremental extension of prior sequence-based meta-learning, and I remain unconvinced that the current theoretical and empirical evidence fully supports the paper’s universality claims, especially for unseen modalities and broader generalization settings. For example, the cross-modal results in Table 4 are not stronger than the Linear Probe. Overall, while the paper has clear merits, the remaining concerns are sufficient for me to stay below the acceptance threshold and keep my original score.

**Key Questions For Authors:**

Please see the weakness.

**Limitations:**

yes

**Strengths And Weaknesses:**

Strengths：

-	The proposed framework is conceptually clean, and the distinction between algorithm-explicit and algorithm-implicit learning is a useful lens for organizing prior and future work.

-	The method is appealing in that it provides a unified way to address varying domains, modalities, and label configurations.

-	The empirical section is reasonably broad and supports the paper’s claim that the method can remain effective beyond standard few-shot settings.


Weaknesses:

-	The proposed method offers limited novelty, as it mainly extends an existing transformer-based sequence meta-learning framework with several additional design choices.

-	The theoretical claims appear stronger than the actual contribution, which seems closer to invariance/equivariance analysis than to a substantive generalization guarantee. And the cross-modal setting still appears to rely heavily on pretrained encoders, making the claimed modality generalization less convincing.

-	The empirical evaluation is limited relative to the breadth of the paper’s universality claims, especially for unseen modalities and broader generalization settings.

-	Experimental evidence for robustness and efficiency is not fully convincing, since some OOD datasets have semantic overlap with the training set, the reported efficiency numbers exclude pretrained encoders.

-	The comparison set is somewhat limited given the scope of the claims, and the paper would benefit from discussion of, and ideally comparison against, more recent related works.

---

> ### Author Rebuttal · Authors · 2026-03-31
>
> Thank you for the detailed assessment, and for recognizing the framework as "clean" and a "useful lens", and the method as "appealing". We address your concerns in detail below.
>
> ---
>
> ### **JWbY.1. Novelty**
>
> > *"The proposed method offers limited novelty, as it mainly extends an existing transformer-based sequence meta-learning framework […]."*
>
> We respectfully disagree. Prior sequence-based meta-learning approaches were limited to toy datasets and single domains.
> To our knowledge, **TAIL is the first model-based meta-learner to succeed simultaneously across domain, modality, and label-space shifts with arbitrary numbers of classes**.
> Several components, as well as their principled integration according to our theoretical framework, are novel:
> - **The formal definitions** of practical universality and algorithm-explicit/implicit taxonomy provides a vocabulary that did not previously exist in the meta-learning literature.
> - **Random injection label embeddings** with a global learnable dictionary are a novel mechanism enabling label space extrapolation. The theoretical analysis is novel as well.
> - **The pricipled combination** of random projections + random injection embeddings + non-causal attention guided by our theory enables simultaneous cross-domain, cross-modal, and label-space-extrapolation generalization in a single model. No existing method achieves this.
>
> These are not minor design choices but innovations that open an entirely new problem setting for meta-learners.
>
> ---
>
> ### **JWbY.2. Theoretical claims and pretrained encoders**
>
> > *"The theoretical claims appear stronger than the actual contribution […]. And the cross-modal setting still appears to rely heavily on pretrained encoders […]."*
>
> We would like to clarify the role of the theoretical contributions. We make **two types** of theoretical contributions:
>
> 1. **The formal framework** (Section 3): Definitions of practical universality, valid learning algorithms, and the explicit/implicit taxonomy. These are definitional contributionsand are not intended as generalization bounds.
> 2. **Analysis of architectural properties** (Section 4.4, Appendix A): These proofs show that TAIL satisfies necessary conditions for practical universality (invariance to feature domain and label encoding). We do not claim it this to be a generalization guarantee. Rather, it proves that TAIL's architecture *does not prevent* generalization across domains and modalities, unlike architectures with fixed feature/label dimensions.
>
> Regarding pretrained encoders: they only provide good representations. Despite **all baselines using the exact same encoders**, GPICL fails on cross-modal transfer, while TAIL succeeds. The meta-learning architecture matters.
> In new experiments with a weaker ResNet-18 encoder (see our response to 3p76.1), TAIL's advantage grows even larger.
>
> ---
>
> ### **JWbY.3. Limited evaluation for unseen modalities**
>
> > *"The empirical evaluation is limited […], especially for unseen modalities and broader generalization settings."*
>
> Our evaluation covers four distinct settings: in-domain, cross-domain, cross-modal, and label space extrapolation (5 to 100-way). This is substantially broader than any prior meta-learning paper we are aware of.
>
> However, we agree that the cross-modal evaluation on a single text dataset is a limitation. To address this, we have conducted additional **audio classification experiments** that we will include in the final revision. Please see our answer to (3p76.2) for a result table and discussion. We will also clarify that TAIL achieves practical universality for **classification**, not for all task types.
>
> ---
>
> ### **JWbY.4. Semantic overlap, efficiency measurement**
>
> > *"some OOD datasets have semantic overlap […]"*
>
> We explicitly acknowledged this and marked datasets with potential overlap with * in Table 3. The three medical datasets were curated to exclude semantic overlap, and TAIL is best or competitive on these. The overlap affects all methods equally.
>
> > *"[…] efficiency numbers exclude pretrained encoders."*
>
> This is intentional and stated in Section 6.2. Since all methods use the same encoders, including them adds the same constant cost and obscures the comparison. The purpose of the efficiency analysis is to compare the meta-learning overhead itself. The result is that GPICL and CAML become infeasible for $K\geq 20$, while TAIL's cost remains comparable to non-meta-learning baselines.
>
> ---
>
> ### **JWbY.5. Comparison set**
>
> > *"The comparison set is somewhat limited […], and the paper would benefit from […] comparison against, more recent related works."*
>
> We compare against the most relevant transformer-based meta-learners (SNAIL, GPICL, CAML) and  non-meta-learning baselines (Linear Probing, ProtoHead). These cover both algorithm-explicit and algorithm-implicit paradigms. If there are **specific methods** you feel would further strengthen this comparison, we would be happy to include them.
>
> ---

---

> > ### Author Rebuttal · Reviewer_JWbY · 2026-04-02
> >
> > Thank you for the detailed rebuttal. It helped clarify several points, and I appreciate the additional discussion. However, as the current submission does not include substantial new experimental evidence, my overall assessment remains unchanged and I will keep my score.

---

> > > ### Author Response · Authors · 2026-04-02
> > >
> > > Thank you for acknowledging the clarifications. We note that you mention follow-up questions and we would be happy to address them and engage in more discussion if you could share them.
> > >
> > > We would like to respectfully point out that we *have* provided new experimental evidence in our rebuttal. While ICML does not permit revised PDF uploads in the rebuttal stage, our rebuttal *does* contain completed experiments with concrete results:
> > >
> > > - **ResNet-18 encoder experiments (table in [answer to 3p76.1](https://openreview.net/forum?id=tAcVihAkb3&noteId=wa9Z9hhLht)):** Evaluation of TAIL with a different image encoder at test time, demonstrating that TAIL's performance does not rely on a specific encoder and that meta-learning gains increase with weaker encoders.
> > > - **Audio classification experiments (table in [answer to 3p76.2](https://openreview.net/forum?id=tAcVihAkb3&noteId=wa9Z9hhLht)):** Evaluation on a third modality (beyond images and text), where TAIL again outperforms all baselines and GPICL collapses to random chance.
> > >
> > > These address the concerns about encoder dependence and the limited cross-modality evaluation. We would also like to point out our ablation experiments already present in the appendix, which provide evidence for the impact of our architectural choices.
> > >
> > > Are there any further experiments or analysis that would help resolve your remaining concerns?

---

### Official Review · Reviewer_R495 · 2026-03-12

**Soundness:** 2
**Presentation:** 3
**Significance:** 3
**Originality:** 2
**Overall Recommendation:** 4
**Confidence:** 4

**Summary:**

This paper proposes a novel concept named universal algorithm-implicit learning. Targeting at more consistent and meaninful definition of universal and general-purpose meta-learning, this work constructs a theorecical framework towards "practical universality" and concept of algorithm-implicit learning. Based on this, a transformer-based algorithm, TAIL, is presented with key advantages in feature encoding, label embedding and efficiency. Experiments on benchmark datasets show the effectiveness of the proposed method.

**Compliance With Llm Reviewing Policy:**

Affirmed.

**Final Justification:**

This paper is well-motivated and clearly presented. The overall algorithm-implicit learning framework is novel and effective. After rebuttal, my main questions have been addressed.

**Key Questions For Authors:**

Please refer to the "weaknesses" part for my initial questions.

**Limitations:**

Yes.

**Strengths And Weaknesses:**

**Strengths**
- This work is well-motivated and clearly presented.
- The proposed method, TAIL, is simple while effective.

**Weaknesses**
- It is claimed that a "theoretical framework" is presented. However,  the framework is build on some definitions, some of which are not sound and clear enough.  Why the expected residual is called "learning curve"? Why valid learning algorithm requires the expected residual monotonically non-increasing in $n$ ? Why "practical universality" closely relates to $n$ instead of the task distribution, as meta-learning does. Once the size of support set $n$ goes to large numbers in order to keep the algorithm valid, is that still practical?
- From my point of view,  "algorithm-implicit" is not that accurate. Given a support set, $\mathcal{A}(S)$ is actually an algrithm. The main difference lies in the support set is considered as an input together with the query. In that way, explicit models/algorithms are absorted into a more universal "implicit" model.
- This work trades large-scale training for generalization across data domains and modality. It is very common for large-scale deep models nowadays. Under this setting, large models trained on large data may have more meta-learning capabilities?
- The experimental comparison with existing meta-learning methods is somewhat not fairly, as TAIL is trained with large-scale data, and the pretrained encoder may further benefit.
- The relationship with in-context learning should be analyzed and compared more deeply and extensively.

*Minor issues*
- Some mathematical notations are not formal enough. E.g., it is not proper to use the size of support set $n$ as the superscipt or lowerscript for support set or distribution. The $a$ in $\alpha_T$ is not defined in Definition 3.4, which is suppose to be $\mathcal{A}$.

---

> ### Author Rebuttal · Authors · 2026-03-31
>
> Thank you for the detailed feedback, and for recognizing the work as well-motivated and clearly presented. We address each concern below and believe that we can fully resolve them.
>
> ---
>
> ### **R495.1. Theoretical framework and definitions**
>
> > *"Why the expected residual is called 'learning curve'?"*
>
> A learning curve describes how the expected performance of an algorithm evolves as a function of the number of training samples.
> The term learning curve is used throughout other works in statistical learning theory with equivalent or similar definitions (see e.g. [1] and [2]).
>
> > *" Why valid learning algorithm requires the expected residual monotonically non-increasing in n?"*
>
> A learning algorithm should, on average, not get *worse* when given more data. An algorithm whose expected error *increases* with additional samples would be counterproductive. This is a deriable and essential property that many classical estimators (eg. k-NN) satisfy.
>
> > *"  Why 'practical universality' closely relates to n instead of the task distribution? Once the size of support set n goes to large numbers […], is that still practical?"*
>
> Practical universality **does** relate to the task distribution. It is defined as **universal validity on a distribution of tasks** that includes test-time tasks with feature and label spaces not represented in the training set. The dependence on $n$ enters through the definition of validity,  which provides finite-sample guarantees (unlike universal consistency, which only makes claims about $n\to\infty$). The definition does not require large $n$; it requires only that for any target error $\varepsilon$, there exists a finite $n$ achieving it. Our experiments evaluate the small-$n$ regime (1-shot, 5-shot).
>
> We will clarify these points in the revision.
>
> [1] Schuurmans, Dale (1997). "Characterizing rational versus exponential learning curves"
> [2] T. Viering and M. Loog (2023), "The Shape of Learning Curves: A Review"
>
> ---
>
> ### **R495.2. "Algorithm-implicit" terminology**
>
> > *"'algorithm-implicit' is not that accurate. Given a support set, A(S) is actually an algorithm."*
>
> We would like to clarify: we do not claim there is *no* algorithm, we say it is **implicit**. It is correct that for any DCI function $g$, there exists an induced algorithm $\mathcal{A}_g$ (stated in Section 3.1). The distinction we make is about **how the algorithm is specified**:
> - **Algorithm-explicit:** The learning procedure $\mathcal{A}$ is defined and specified beforehand (e.g., "run $k$ steps of gradient descent).
> - **Algorithm-implicit:** No learning procedure is specified. The behavior of $g(S, \cdot)$ *implies* an algorithm $\mathcal{A}_g$, but this algorithm is never explicitly defined. It emerges from the training.
> The observation that "explicit algorithms are absorbed into a more universal implicit model" is consistent with our framework.
>
> ---
>
> ### **R495.3. Large-scale training**
>
> > *"This work trades large-scale training for generalization across data domains and modality."*
>
> We agree that using large-scale training to enable generalization is a common theme in modern deep learning and we explicitly draw this analogy in Section 1. However, our contribution is not simply "train on more data." It is designing an architecture with specific structural properties (random projections, random injection label embeddings, non-causal attention) to on the one hand **exploit** diverse training data and on the other hand **generalize** across domains and modalities. The baselines were trained on the **exact same large-scale data** and failed to achieve comparable generalization, demonstrating that architecture design matters beyond data scale.
>
> If the question relates to foundation models, see also our answer to (DhcM.3): Our approach is complementary to and can benefit from better foundation models. We provide a single forward-pass inference model that generalizes across modalities as an alternative to fine-tuning/prompting large models.
>
> ---
>
> ### **R495.4. Fairness of experimental comparison**
>
>
> > *"The experimental comparison with existing meta-learning methods is somewhat not fair, as TAIL is trained with large-scale data, and the pretrained encoder may further benefit."*
>
> All methods use the exact same pretrained encoders and the exact same meta-training data, which we have stated in Section 6.1: *"For all baseline methods, as well as TAIL (ours), we use fixed pretrained backbones as feature encoders. To ensure a fair comparison, all meta-learning algorithms were trained on the same meta-data and used the same pretrained encoders."*
>
> The encoders are frozen for all methods. The **only** difference between methods is the meta-learning architecture, which is exactly what we aim to evaluate. Since we agree that this is a crucial point regarding the fairness of the evaluation, we will make this point more prominent in the camera-ready version.

---

> > ### Author Rebuttal · Reviewer_R495 · 2026-04-04
> >
> > Thanks for the rebuttal. My main concerns have been addressed. I will rate the score accordingly.

---

### Official Review · Reviewer_DhcM · 2026-03-13

**Soundness:** 3
**Presentation:** 3
**Significance:** 3
**Originality:** 3
**Overall Recommendation:** 4
**Confidence:** 2

**Summary:**

This paper proposes TAIL, a transformer-based algorithm-implicit meta-learner designed for few-shot classification under heterogeneous settings, including varying feature spaces, varying label spaces, and cross-domain or cross-modal transfer. The paper introduces a framing of algorithm-explicit vs. algorithm-implicit learning and argues that the latter is better suited to broad few-shot generalization. To support this, TAIL combines modality-specific pretrained encoders, projection into a shared feature space, a flexible global label-embedding mechanism, and non-causal transformer inference over support and query examples. Empirically, the paper shows strong results on standard in-domain benchmarks, cross-domain image transfer, image-to-text transfer, and extrapolation to larger test-time label spaces than seen during training. Overall, the paper is primarily a method-and-evaluation paper with a conceptual framing contribution, and its main value lies in pushing meta-learning toward broader classification generalization rather than proposing a universal learner in a fully general sense.

**Compliance With Llm Reviewing Policy:**

Affirmed.

**Final Justification:**

The rebuttal addressed my main concerns.

**Key Questions For Authors:**

1. How broadly should readers interpret the “practical universality” claim?
   The experiments are strong, but they are still centered on few-shot classification. Could the authors clarify the intended scope of this claim and how far they believe it extends beyond classification tasks?

2. How robust is the cross-modal generalization claim beyond the current text benchmark?
   The image-to-text transfer result is interesting, but it is still a limited text classification setting. Could the authors comment on how broadly they expect this result to hold across other text tasks?

**Limitations:**

Np.

**Strengths And Weaknesses:**

* Soundness.

  The paper studies few-shot generalization and presents a coherent method, TAIL, with clear design choices for handling variable feature spaces, variable label spaces, and efficient transformer-based inference. The empirical study is fairly broad by meta-learning standards, covering in-domain, cross-domain, cross-modal, and label-space extrapolation settings, and the ablations are useful.

  The main weakness is that the claim, moving toward practical universality, is supported only in a fairly narrow problem family, namely few-shot classification. The evidence shows strong generalization within that setting, but not yet a broader universal learner in the sense readers might infer.

* Presentation.

  The paper is generally well organized and the high-level ideas are interesting, especially the framing of algorithm-explicit vs. algorithm-implicit learning.

* Significance.

  This is a meaningful paper within meta-learning. It pushes beyond the standard fixed-way, fixed-domain setting and shows unusually strong cross-domain, cross-modal, and label-space transfer for an episodic meta-learner. That is a real contribution.

  However, the broader significance is less clear in today’s research landscape. Much of current few-shot progress is increasingly driven by foundation-model adaptation, prompting, and in-context learning, rather than standalone meta-learners. Against that trend, this work feels more like a strong advance inside meta-learning than a result likely to reshape the larger few-shot learning agenda.

* Originality.

  The paper is reasonably original in both framing and method. The distinction between algorithm-explicit and algorithm-implicit learners is a useful perspective, and TAIL combines universal feature handling, universal label handling, and transformer-based episodic inference in a novel way.

---

> ### Author Rebuttal · Authors · 2026-03-31
>
> Thank you for your thoughtful evaluation, and for recognizing that TAIL pushes beyond the standard fixed-way, fixed-domain setting with unusually strong cross-domain, cross-modal, and label-space transfer. We address your questions in detail below.
>
>
> ---
>
>
> ### **DhcM.1. Scope of the "practical universality" claim**
>
>
> > *"How broadly should readers interpret the 'practical universality' claim? The experiments are strong, but they are still centered on few-shot classification. Could the authors clarify the intended scope of this claim and how far they believe it extends beyond classification tasks?"*
>
>
> We appreciate this question and want to be precise. Practical universality, as we formally define it (Section 3.3), is a property of a learning algorithm: it requires validity across tasks with varying feature domains, label domains, and data distributions. Our **theoretical framework** is general and not restricted to classification. Definitions 3.2–3.5 apply to any supervised learning task.
>
> However, **TAIL as a method** is designed for and evaluated on classification. We believe this is appropriate as a first demonstration: classification with varying label spaces, domains, and modalities already represents a substantial advance over prior work, which is typically restricted to fixed-way, fixed-domain settings. We do not claim that TAIL is a universal learner in the broadest sense. Rather, we claim it achieves practical universality *within classification* and makes advances towards practical universality in general (as stated in the introduction).
>
> We will add clarifying language in the camera-ready version to explicitly state that the theoretical framework is task-type-agnostic, that TAIL's current empirical validation covers only classification, and that extending to regression, structured prediction, and other task types will be considered in future work.
>
>
> ---
>
>
> ### **DhcM.2. Robustness of cross-modal generalization beyond the current text benchmark**
>
>
> > *"The image-to-text transfer result is interesting, but it is still a limited text classification setting. Could the authors comment on how broadly they expect this result to hold across other text tasks?"*
>
>
> If the question concerns other **text classification** tasks: we expect the result to hold, since TAIL is agnostic to the label domain of the task by construction. The only requirement is a reasonable pretrained encoder for the given language.
> To provide concrete evidence beyond the text benchmark, we have conducted additional experiments on **audio classification** (music genre classification). The results are consistent: TAIL achieves 55.33% / 40.63% (5-shot / 1-shot), outperforming all baselines, while GPICL again collapses to random chance (20.03%). This confirms that the cross-modal transfer is not specific to the text setting.
>
> If the question concerns **generative or non-classification** tasks: this goes beyond the current scope of TAIL, which is designed as a universal classifier. However, the theoretical framework is task-type-agnostic and TAIL's core mechanisms (random projection into shared latent space, non-causal transformer) do not inherently assume classification. In principle, replacing the classification head with a regression head or a structured output head could extend the approach to other task types. We view this as a promising direction for future work and will discuss it in the revision.
>
>
> ---
>
>
> ### **DhcM.3. Significance relative to foundation model adaptation**
>
>
> > *"Much of current few-shot progress is increasingly driven by foundation-model adaptation, prompting, and in-context learning, rather than standalone meta-learners."*
>
>
> This is a fair observation. We see TAIL as **complementary** to the foundation-model trend rather than competing with it. TAIL can directly benefit from improvements in foundation models, since it uses pretrained encoders as a component. The two paradigms are not mutually exclusive.
>
> Foundation model fine-tuning requires retraining for each new task, which is computationally expensive when tasks arrive frequently. Prompt-based methods require task-specific prompt design and are typically restricted to a single modality. In contrast, TAIL provides single-forward-pass inference with no test-time training loop and generalizes across modalities.
>
> We will add the discussion of this relationship to the paper.
>
>
>
> ---

---

> > ### Author Rebuttal · Reviewer_DhcM · 2026-04-04
> >
> > Thank you for the detailed reply. My concerns are addressed.

---

### Official Review · Reviewer_3p76 · 2026-03-15

**Soundness:** 3
**Presentation:** 3
**Significance:** 3
**Originality:** 3
**Overall Recommendation:** 5
**Confidence:** 4

**Summary:**

This paper studies universal meta-learning, and introduces TAIL (Transformer-based Algorithm-Implicit Learner), a novel meta-learning framework designed to achieve practical universality (the ability to learn across tasks with differing feature domains, modalities, and label spaces) which is also formalized in the paper and this paper also develops a theoretical framework distinguishing algorithm-explicit and algorithm-implicit meta-learners.

Extensive experiments validate the effectiveness of TAIL including cross-domain, cross-modality generalization, and demonstrate robustness to unseen tasks with up to 20× more classes than in training.

**Compliance With Llm Reviewing Policy:**

Affirmed.

**Key Questions For Authors:**

1. Since all methods rely on pre-trained encoders, could the authors provide analysis to clarify how much of the cross-domain/modality performance comes from the meta-learning algorithm versus pre-trained encoders?

2. Experiments on cross-modality evaluate transfer from images to text classification on a single dataset. Have the authors tested the method on additional modalities or tasks to further validate the universality claim?

3. It seems only comparison experiments with and without $\pi$ were conducted. How sensitive is the model to the dimensionality and randomness of the projection operator used for universal feature encoding? Compared to the results difference (table 5), it looks the sensitivity is not very significant?

**Limitations:**

The paper should also discuss the limitations of this work. Some could be (kind of consistent with the questions listed above):
1. How much knowledge is meta-learned through the proposed meta-learning structure, not the pretrained model?
2. Only one experiment on a single image to text classification dateset to validate the ability of cross modality. Should we also need to contain more experiments across more datasets and more modalities?

**Strengths And Weaknesses:**

Soundness: The paper is generally technically sound.

Presentation: The paper is clearly written and well organized.

Significance: The goal of designing meta-learning systems that generalize across heterogeneous (even unseen) tasks and modalities is an important research direction. The proposed perspective on algorithm-implicit learning and the demonstrated cross-modality results may stimulate further work on universal meta-learning systems.

Originality: The main novelty lies in the conceptual framing of algorithm-implicit learning and the integration of transformers, random projections, and randomized label embeddings within a unified meta-learning architecture. While the components themselves are largely known, the combination provides a useful perspective on designing more general meta-learning models.

---

> ### Author Rebuttal · Authors · 2026-03-31
>
> Thank you for your time and positive assessment of our work. We are grateful for the constructive suggestions and address them in detail below.
>
> ---
>
> ### **3p76.1. Pretrained encoder vs. meta-learning contribution**
>
> > *"[C]ould the authors provide analysis to clarify how much of the cross-domain/modality performance comes from the meta-learning algorithm versus pre-trained encoders?"*
>
> Thank you for raising this. We would like to clarify that **TAIL and all baselines already use the exact same pretrained encoders** (ViT-H/OpenCLIP for vision, DistilBERT for text), and these encoders are frozen for all methods. Therefore, all performance differences are **entirely attributable to the meta-learning algorithm**. This is most visible when comparing TAIL to GPICL on cross-modality (Table 4): both use the same encoders and random projections, yet GPICL collapses to random chance.
> Linear Probing / ProtoHead use the pretrained encoder with no meta-learned component and represent what the encoder alone can provide. TAIL consistently outperforms both, demonstrating that the meta-learned algorithm adds value beyond the encoder representations.
>
> **New experiment: ResNet-18 encoder.** To further isolate the meta-learning algorithm, we additionally ran experiments replacing the ViT-H encoder with a substantially weaker ResNet-18 (pretrained on ImageNet-1k) only at test time. TAIL was still meta-trained with ViT-H features. This tests how well the meta-learned algorithm transfers when the underlying representations change drastically. The results below show that TAIL's advantage over the encoder-only baselines is **maintained and in fact becomes even larger** with a weaker encoder.
>
> **Average in-domain and cross-domain results with ResNet-18 encoder (mean accuracy % over 1000 episodes):**
> | | In-domain avg. (5-shot) | Cross-domain avg. (5-shot) | In-domain avg. (1-shot) | Cross-domain avg. (1-shot) |
> |---|---|---|---|---|
> | Linear Probe | 76.25 | 54.20 | 57.07 | 40.33 |
> | ProtoHead | 87.59 | 63.67 | 70.58 | 46.49 |
> | **TAIL (ours)** | **88.91** | **66.08** | **79.46** | **53.32** |
>
> **Summary:** With the weaker ResNet-18 encoder, the performance of all methods drops compared to ViT-H. However TAIL consistently and significantly outperforms Linear Probing and ProtoHead. Notably, TAIL's advantage is **larger** with ResNet-18 than with ViT-H, demonstrating that the meta-learning algorithm contributes significantly beyond the encoder, and contributes *more* when the encoder is weaker. The full per-dataset result tables will be included in the appendix of the camera-ready version.
>
> ---
>
> ### **3p76.2. Additional modalities for cross-modal evaluation**
>
> > *"Have the authors tested the method on additional modalities or tasks to further validate the universality claim?"*
>
> We agree that results on additional modalities would strengthen the universality claim and have therefore performed additional experiments on a **audio classification** dataset. We will incorporate these results in the camera-ready revision. The results on music genre classification paint a similar picture as the text classification results, with TAIL outperforming the baselines consistently:
>
> | Method | 5-shot | 1-shot |
> |--------|--------|--------|
> | Linear Probe | 38.02 ± 0.57 | 29.58 ± 0.51 |
> | ProtoHead | 54.74 ± 0.58 | 39.09 ± 0.63 |
> | GPICL | 20.03 ± 0.08 | 19.98 ± 0.13 |
> | **TAIL (ours)** | **55.33 ± 0.60** | **40.63 ± 0.62** |
>
> ---
>
> ### **3p76.3. Sensitivity to the projection operator π**
>
> > *"How sensitive is the model to the dimensionality and randomness of the projection operator[…]? Compared to the results difference (table 5), it looks the sensitivity is not very significant?"*
>
> We want to start by clarifying that $\pi$ is **re-sampled for every episode**, both during training and evaluation. The model never relies on any particular projection but instead learns to be invariant to the specific $\pi$ (which is formally proven in Theorem A.9). This randomization helps cross-domain and cross-modal generalization.
>
> As you observed, the differences in Table 5 are moderate for in-domain settings. However, the effect becomes substantial in the cross-modality setting (increase from 66.69 to 84.87%, 1-shot). The random projection mechanism is not critical for in-domain performance, but is essential for the generalization necessary for practical universality.
>
> Regarding dimensionality: the projection target dimension $d_{\text{data}}$ is determined by the transformer's hidden dimension and the source dimension is determined by the encoder output, so these dimensions are set by architectural choices rather than being free hyperparameters to tune.
>
> ---
>
> ### **3p76.4. Limitations**
>
> We agree and will add a discussion of the following limitations:
> - The practical universality claim for TAIL is currently limited to classification.
> - Albeit less so than other methods, TAIL still relies on the quality of the encoder for SOTA results.
>
> ---

---

> > ### Author Rebuttal · Reviewer_3p76 · 2026-04-06
> >
> > The response has resolved my concerns.

---

### Decision · Program_Chairs · 2026-04-30

**Decision:**

Accept (regular)

**Comment:**

After carefully reviewing the manuscript, the reviewers' comments, and the authors' rebuttal, I find that the paper demonstrates solid novelty and a clear, well-motivated research direction. Reviewer JWbY raised valid concerns, and while the authors provided some clarifications in their response, I believe certain points have not been fully resolved. Overall, I recommend acceptance of this paper, with the expectation that the authors will thoroughly address the remaining issues and carefully revise the manuscript for the camera-ready version.